# Prefrontal cortex supports speech perception in listeners with cochlear implants

**Arefeh Sherafati[1], Noel Dwyer[2], Aahana Bajracharya[2], Mahlega Samira Hassanpour[3], Adam T Eggebrecht[1,4,5,6], Jill B Firszt[2], Joseph P Culver[1,5,6,7], Jonathan E Peelle[2]\***

[1]Department of Radiology, Washington University in St. Louis, St. Louis, United States; [2]Department of Otolaryngology, Washington University in St. Louis, St. Louis, United States; [3]Moran Eye Center, University of Utah, Salt Lake City, United States; [4]Department of Electrical & Systems Engineering, Washington University in St. Louis, St. Louis, United States; [5]Department of Biomedical Engineering, Washington University in St. Louis, St. Louis, United States; [6]Division of Biology and Biomedical Sciences, Washington University in St. Louis, St. Louis, United States; [7]Department of Physics, Washington University in St. Louis, St. Louis, United States

**\*For correspondence:**
j.peelle@northeastern.edu

**Abstract** Cochlear implants are neuroprosthetic devices that can restore hearing in people with severe to profound hearing loss by electrically stimulating the auditory nerve. Because of physical limitations on the precision of this stimulation, the acoustic information delivered by a cochlear implant does not convey the same level of acoustic detail as that conveyed by normal hearing. As a result, speech understanding in listeners with cochlear implants is typically poorer and more effortful than in listeners with normal hearing. The brain networks supporting speech understanding in listeners with cochlear implants are not well understood, partly due to difficulties obtaining functional neuroimaging data in this population. In the current study, we assessed the brain regions supporting spoken word understanding in adult listeners with right unilateral cochlear implants (n=20) and matched controls (n=18) using high-density diffuse optical tomography (HD-DOT), a quiet and non-invasive imaging modality with spatial resolution comparable to that of functional MRI. We found that while listening to spoken words in quiet, listeners with cochlear implants showed greater activity in the left prefrontal cortex than listeners with normal hearing, specifically in a region engaged in a separate spatial working memory task. These results suggest that listeners with cochlear implants require greater cognitive processing during speech understanding than listeners with normal hearing, supported by compensatory recruitment of the left prefrontal cortex.

## Editor's evaluation

The work establishes use of a specific extra area of prefrontal cortex during word listening by CI users and supports a hypothesis based on the multiple demand network that can be tested using other techniques that look at the rest of the network. The revision provides further points of clarity required and better acknowledges the limitations of the technique.

## Introduction

Cochlear implants (CIs) are neuroprosthetic devices that can restore hearing in people with severe to profound hearing loss by electrically stimulating the auditory nerve. Because of physical limitations

on the precision of this stimulation—including, for example, the spatial spread of electrical current (*Garcia et al., 2021*)—the auditory stimulation delivered by a CI does not convey the same level of acoustic detail as normal hearing. As a result, speech understanding in listeners with CIs is poorer than in listeners with normal hearing (*Firszt et al., 2004*). Notably, even in quiet, listeners with CIs report increased effort during listening (*Dwyer and Firszt, 2014*). Despite these challenges, many listeners with CIs attain significant success in understanding auditory speech. This remarkable success raises the question of how listeners with CIs make sense of a degraded acoustic signal.

One area of key importance is understanding the degree to which listeners with CIs rely on non-linguistic cognitive mechanisms to compensate for a degraded acoustic signal. In listeners with normal hearing, cognitive demands increase when speech is acoustically challenging (*Peelle, 2018*). For example, even when speech is completely intelligible, acoustically degraded speech can reduce later memory for what has been heard (*Cousins et al., 2013*; *Koeritzer et al., 2018*; *Rabbitt, 1968*; *Ward et al., 2016*). These findings suggest that to understand acoustically challenging speech, listeners need to engage domain-general cognitive resources during perception. In a limited capacity cognitive system (*Wingfield, 2016*), such recruitment necessarily reduces the resources available for other tasks, including memory encoding. Importantly, even speech presented in quiet (i.e., without background noise) is degraded by the time it reaches the auditory system of a listener with a CI.

Cognitive demands during speech understanding are supported by several brain networks that supplement classic frontotemporal language regions. The cingulo-opercular network, for example, is engaged during particularly challenging speech (*Eckert et al., 2009*; *Vaden et al., 2017*) and supports successful comprehension during difficult listening (*Vaden et al., 2013*). Recruitment of prefrontal cortex (PFC) complements that in the cingulo-opercular network and varies parametrically with speech intelligibility (*Davis and Johnsrude, 2003*). Activity in PFC, particularly dorsolateral regions, is associated with cognitive demands in a wide range of tasks (*Duncan, 2010*), consistent with domain-general cognitive control (*Braver, 2012*). We thus hypothesized that listeners with CIs would rely more on PFC during listening than listeners with normal hearing, and in particular regions of PFC associated with non-linguistic tasks. However, the functional anatomy of PFC is also complex (*Noyce et al., 2017*), and dissociating nearby language and domain-general processing regions is challenging (*Fedorenko et al., 2012*).

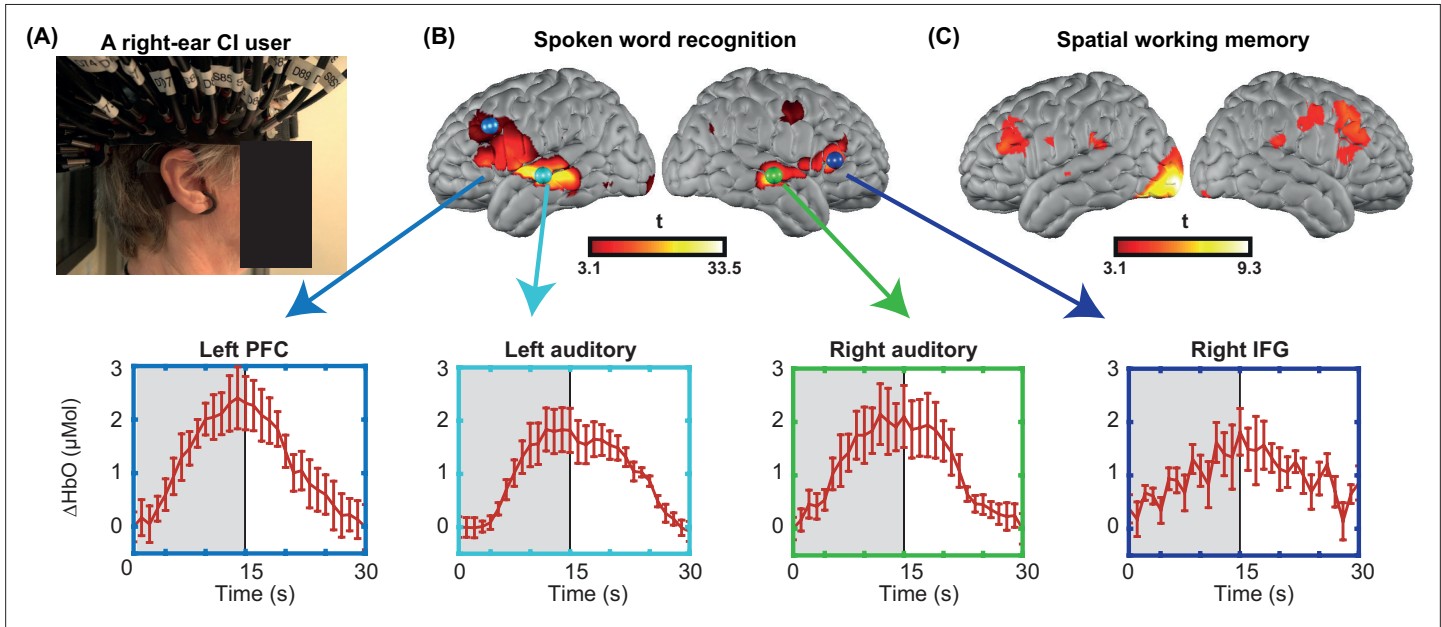

**Figure 1.** Single-subject data from one cochlear implant (CI) user across multiple sessions. (**A**) A CI user wearing the high-density diffuse optical tomography (HD-DOT) cap. (**B**) Response to the spoken words across six sessions (36 min of data). Hemodynamic response time-traces are plotted for peak activation values across six sessions for four brain regions. The seed colors match the plot boundaries with error bars indicating the standard error of the mean over n=12 runs of data. Gray shaded region indicates period during which words are presented. (**C**) Response to the spatial working memory task for the same CI user across four sessions (32 min of data).

A central question concerns the degree to which listeners with CIs rely on cognitive processing outside core speech regions, such as dorsolateral PFC. Obtaining precise spatially localized images of regional brain activity has been difficult in listeners with CIs, given that functional MRI (fMRI) is not possible (or subject to artifact) due to the CI hardware. Thus, optical brain imaging (*Peelle, 2017*) has become a method of choice for studying functional activity in CI listeners (*Anderson et al., 2017*; *Lawler et al., 2015*; *Lawrence et al., 2018*; *Olds et al., 2016*; *Zhou et al., 2018*). In the current study, we use high-density diffuse optical tomography (HD-DOT) (*Eggebrecht et al., 2014*; *Zeff et al., 2007*), previously validated in speech studies in listeners with normal hearing (*Hassanpour et al., 2015*; *Hassanpour et al., 2017*; *Schroeder et al., 2020*). HD-DOT provides high spatial resolution and homogenous sensitivity over a field of view that captures known speech-related brain regions (*White and Culver, 2010*). We examine the brain regions supporting single word processing in listeners with a right unilateral CI relative to that in a group of matched, normal-hearing controls. We hypothesized that listeners with CIs would exhibit greater recruitment of PFC compared to normal-hearing controls.

## Results
### Multi-session single-subject results
Due to the variability across CI users and difficulties in defining single-subject regions of interest (ROIs), we performed a small multi-session study from one CI subject for six sessions (*Figure 1*). We collected two runs of spoken word perception per session (for six sessions) and one run of spatial working memory task per session (for four sessions). This multi-session analysis enabled localizing the

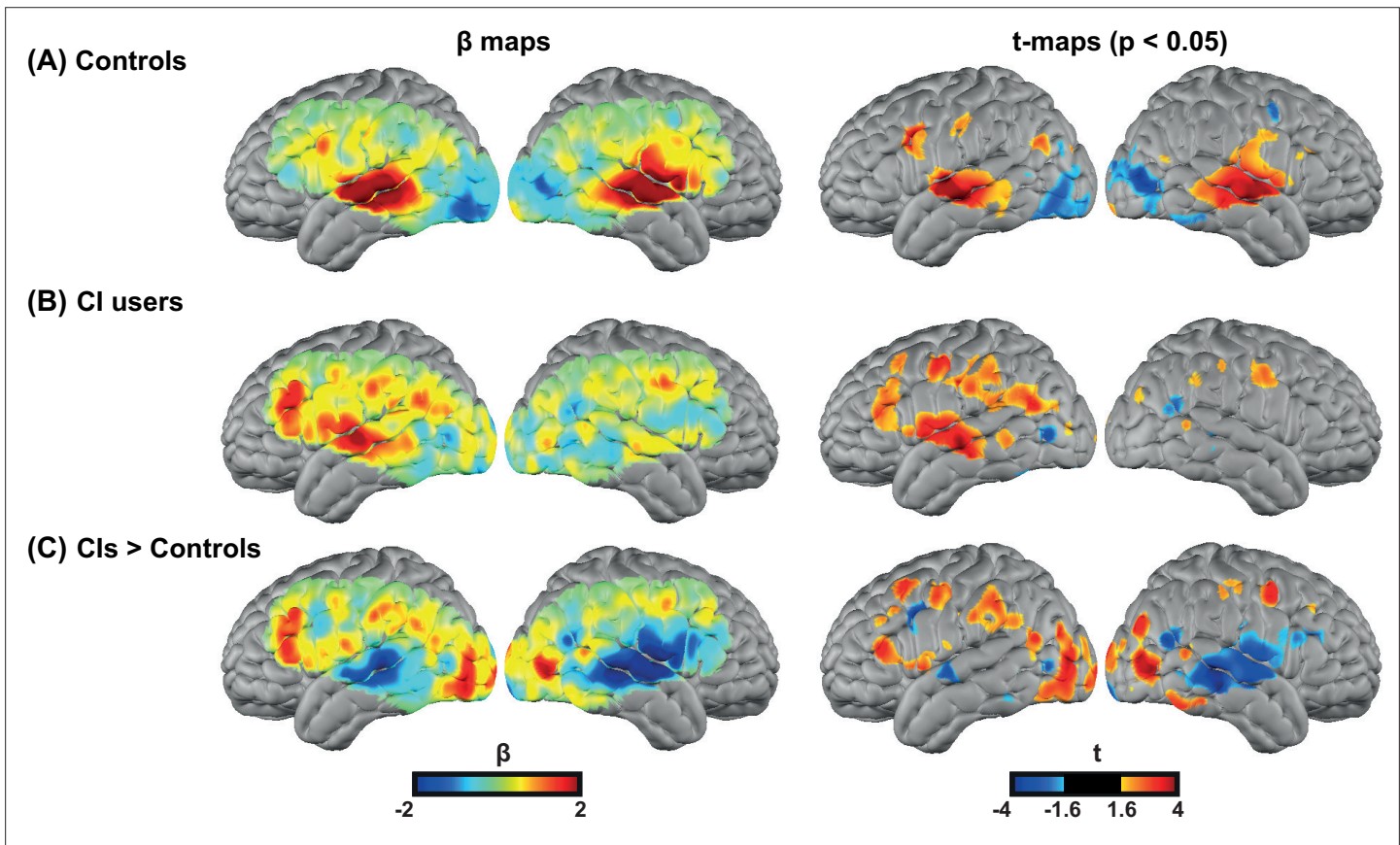

**Figure 2.** Spoken word recognition group maps. Response to the spoken words in (**A**) 18 controls and (**B**) 20 right ear cochlear implant (CI) users. (**C**) Differential activation in response to the spoken words task in CIs>controls highlights the group differences. The first column shows unthresholded β maps and the second column shows t-statistic maps thresholded at voxelwise p<0.05 (uncorrected) for each group.

The online version of this article includes the following figure supplement(s) for figure 2:

**Figure supplement 1.** Group results for different hemoglobin contrasts.

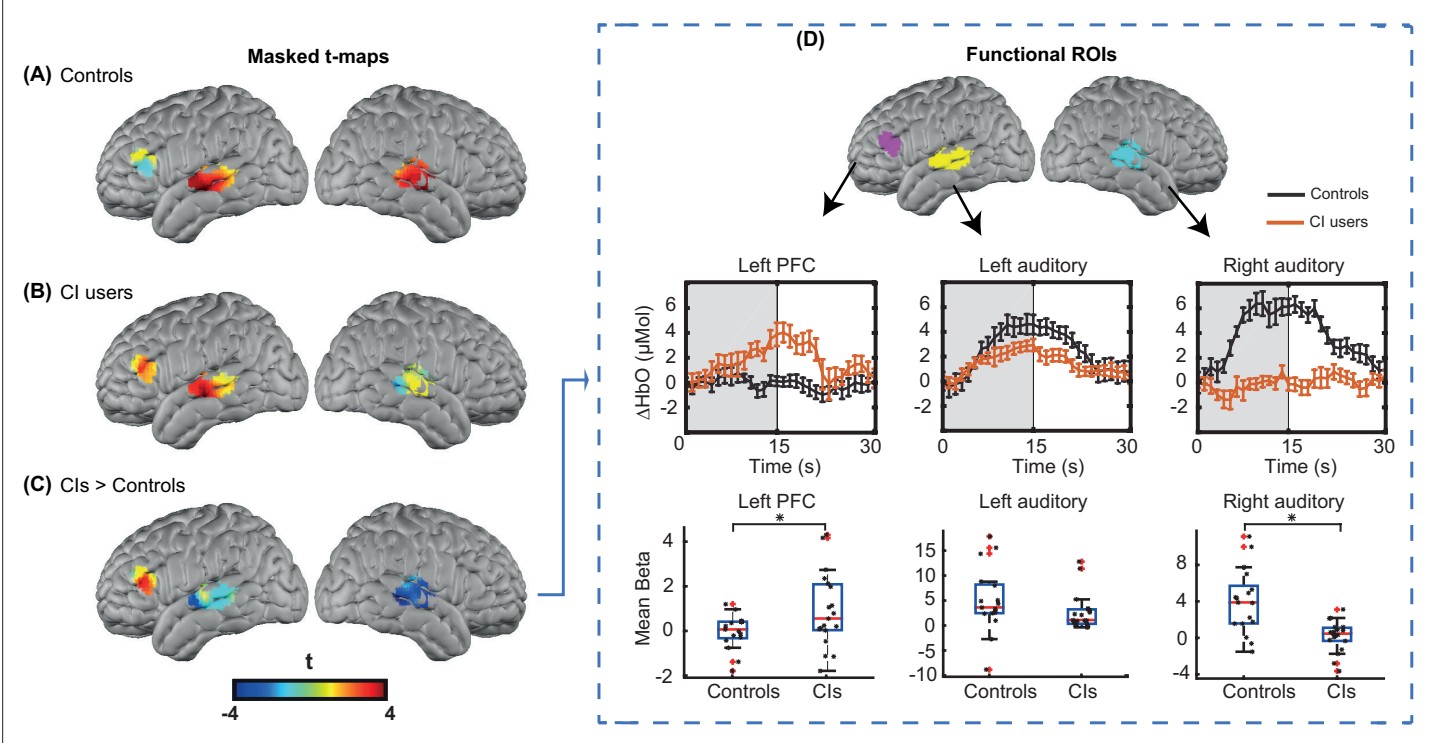

**Figure 3.** Region of interest (ROI)-based statistical analysis for spoken word recognition task. Unthresholded t-maps in response to the spoken words spatially masked in the three ROIs for (**A**) 18 controls, (**B**) 20 right ear cochlear implant (CI) users, and (**C**) CIs>controls, highlight the group differences in certain brain areas. (**D**) Temporal profile of the hemodynamic response in three selected ROIs (left prefrontal cortex [PFC], left auditory, and right auditory cortices). The error bars indicate the standard error of the mean over n=20 for CI users and n=18 for controls. Two-sample t-tests for mean β value in each ROI have been calculated between controls and the CI user group, confirming a significant increase in left PFC (p=0.015) and a significant decrease in the right auditory cortex (p=0.0017) in CI users (indicated by an asterisk above their corresponding box plots, corrected for multiple comparisons). The observed change in the left auditory cortex was not significant (p=0.15).

The online version of this article includes the following source data and figure supplement(s) for figure 3:

**Figure supplement 1.** Coupling coefficients of each source and detector is shown in a flat view for all cochlear implant (CI) users included in the study.

**Figure supplement 2.** Coupling coefficients of each source and detector is shown in a flat view for all controls included in the study.

**Figure supplement 3.** Effect of the simulated cochlear implant (CI) transducer in controls in the right auditory region of interest (ROI) analysis.

**Source data 1.** Noisy source-detector numbers are provided along with the threshold used for identifying them for each cochlear implant (CI) user.

left and right PFC based on the non-verbal spatial working memory task for this subject (*Figure 1C*). It also revealed the engagement of regions beyond the auditory cortex, including PFC, during the word perception task (*Figure 1B*). Time-traces of oxyhemoglobin concentration change show a clear event-related response for four selected regions in the word perception results.

## Mapping the brain response to spoken words

We first investigated the degree of auditory activation in both control and CI groups by assessing the activity in a block design single word presentation task. We found strong bilateral superior temporal gyrus (STG) activations in controls similar to our previous studies using the same paradigm (*Eggebrecht et al., 2014*; *Sherafati et al., 2020*), as well as a strong left STG and a reduced right STG activation for the CI users (*Figure 2A–B*). In addition, we observed strong left-lateralized activations in regions beyond the auditory cortex, including parts of the PFC, in the CI group (*Figure 2B*).

*Figure 2—figure supplement 1* provides the β maps of oxyhemoglobin (HbO), deoxyhemoglobin (HbR), and total hemoglobin (HbT) for controls (panel A), CI users (panel B), CIs>controls (panel C), and controls>CIs (panel D).

For statistical analysis, we focused on our predefined ROIs, as shown in Figure 6. *Figure 3A-C* shows unthresholded t-maps, masked by our ROIs. We averaged β values within each ROI, and statistically tested for group differences correcting for multiple comparisons across the three ROIs

(*Figure 3D*). The temporal profile of the hemodynamic response in three selected ROIs also suggests the increased activity in the left PFC in the CI users relative to controls, and a decreased activity in both left and right auditory regions. Two sample t-statistics for the mean β values in each ROI support a statistically significant difference between the control and CI groups in left PFC, t(27) = 2.3, p=0.015 (one-tailed) and right auditory cortex, t(23) = 3.54, p=0.0017 (two-tailed) (*Figure 3D*). The observed change in the left auditory cortex was not statistically significant, t(36) = 1.46, p=0.15 (two-tailed). The threshold for statistical significance, Bonferroni corrected for multiple comparisons across three ROI analyses, was 0.016.

To further investigate whether the significant decrease in the right auditory cortex might be due to lower light level values around the CI transducer, we first performed a simulation of the HD-DOT sensitivity profile by blocking the optodes around the CI transducer. We found minimal overlap between the CI-related signal loss and the right auditory ROI (Appendix 2). We also performed an additional simulation analysis by first calculating the mean value of optical light power for each measurement. Then, for each source-detector location, we defined the approximate coupling coefficient as the mean over the first nearest neighbor measurements that have that source or detector in common. *Figure 3—figure supplements 1–2* show the coupling coefficients of each source and detector for all CI users and controls in the flat view, corrected by the average sensitivity of the avalanche photodiode detectors (APDs) plotted on a log base 10 scale to create linearly distributed color scale. We defined noisy optodes in the CI cohort as ones that had a coupling coefficient of 30% of maximum or lower based on the values for each participant across their two speech runs. If no optode was identified using this threshold, we lowered the threshold by 10% increments until at least one source or detector was identified (under the logic that the CI transducer must block some light). *Figure 3—source data 1* shows the source-detector numbers and the threshold used for identifying them for each CI user. For controls, no source or detector had a coupling coefficient of less than 30% of the maximum.

After identifying the noisy source-detector numbers for all 20 CI users, we pre-processed the data for 18 controls 100 times. For each of these analyses, we blocked the identified sources and detectors in a selected CI user for one control, assigned by a random permutation, effectively simulating the CI-based dropout in our control participants. Then, we replicated the ROI analysis for the right auditory cortex in *Figure 3*, far right panel, and found that in all 100 shuffles, the right auditory cortex in control subjects still had a significantly larger activation compared to the CI users (*Figure 3—figure supplement 3*).

## Behavioral measures

An important consideration in studying CI users is the variability in their speech perception abilities, hearing thresholds, and the relationship with brain activity. *Figure 4* shows exploratory analyses between the magnitude of the activation in the left PFC ROI for the CI cohort with respect to the

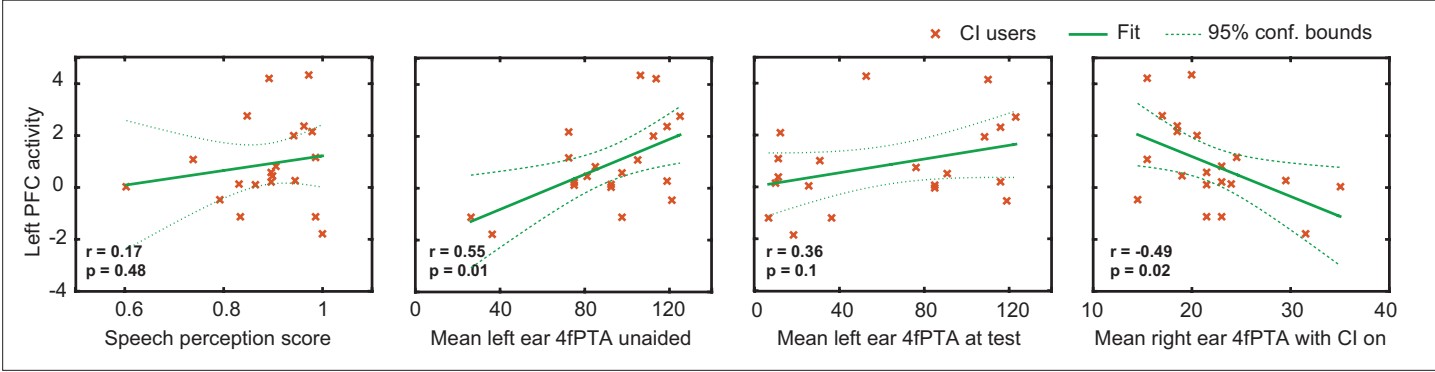

**Figure 4.** Relationship between the magnitude of activation in left prefrontal cortex (PFC) and behavioral scores in cochlear implant (CI) users. Plots of the Pearson correlation r between the magnitude of the mean β value in the left PFC region of interest (ROI) are shown with respect to speech perception score, left ear hearing threshold unaided, left ear hearing threshold (aided if the subject used a hearing aid), and right ear CI-aided hearing threshold. Hearing threshold was defined as four-frequency pure-tone average (4fPTA) at four frequencies, 500, 1000, 2000, and 4000 Hz.

The online version of this article includes the following figure supplement(s) for figure 4:

**Figure supplement 1.** Audiograms for left and right ears.

speech perception score, left ear hearing threshold unaided, left ear hearing threshold at test (aided if the subject used a hearing aid), and right ear hearing threshold (CI-aided).

Using p<0.05 (uncorrected) as a statistical threshold, left PFC activation positively correlated with left ear unaided thresholds (p=0.01, Pearson r=0.55) and negatively correlated with right ear CI-aided thresholds (p=0.02, Pearson r=–0.49). Left PFC activation did not correlate with speech perception score (p=0.4, Pearson r=0.17) and aided hearing threshold for the left ear (p=0.1, Pearson r=0.36). *Figure 4—figure supplement 1* shows the hearing thresholds (audiograms) for left and right ears for each control and CI participant. These scores were unavailable for one control participant, who was thus not included in these analyses.

## Discussion

Using high-density optical brain imaging, we examined the brain networks supporting spoken word recognition in listeners with CIs relative to a matched group of controls with bilateral normal hearing. We found that relative to controls, when listening to words in quiet, listeners with CIs showed reduced activity in the right auditory cortex and—critically—increased activity in left PFC. We review these two findings in turn below.

### Increased PFC activity in CI users

When listening to spoken words in quiet, listeners with normal hearing typically engage the left and right superior temporal cortex, including primary and secondary auditory regions (*Binder et al., 2000*; *Price et al., 1992*; *Rogers et al., 2020*; *Wiggins et al., 2016*). Our current results for controls show this same pattern. However, when listeners with CIs performed the same task, we found that they also engaged left PFC significantly more than the controls.

Although we only tested a single level of speech difficulty (i.e., speech in quiet), prior studies have parametrically varied speech intelligibility and found intelligibility-dependent responses in the PFC. Use of several types of signal degradation (*Davis and Johnsrude, 2003*) revealed a classic 'inverted-U' shape response in the PFC as a function of speech intelligibility, with activity increasing until the speech became very challenging and then tapering off. A similar pattern was reported in functional near-infrared spectroscopy (fNIRS) (*Lawrence et al., 2018*).

A pervasive challenge for understanding the role of PFC in speech understanding is the close anatomical relationship of core language processing regions and domain-general regions of PFC (*Fedorenko et al., 2012*). We attempted to add some degree of functional specificity to our interpretation by including a spatial working memory task presumed to strongly engage domain-general regions with minimal reliance on language processing (*Duncan, 2010*; *Woolgar et al., 2015*). Ideally, we would have used functional ROIs individually created for each subject. However, we were not convinced that our data for this task were sufficiently reliable at the single-subject level, as we only had one run per subject. Furthermore, we did not have spatial working memory task data for all subjects. Thus, our functional localization relies on group-average spatial working memory responses. The region we identified—centered in left inferior frontal sulcus—corresponds well to other investigations of non-language tasks (e.g., *Duncan, 2010*), and supports our preferred interpretation of engagement of domain-general regions of dorsolateral PFC during listening. However, the ROI also extends into the dorsal portion of inferior frontal gyrus (IFG), and we cannot rule out the possibility that this frontal activity relates to increased language (as opposed to domain-general) processing.

### Reduced auditory cortical activity in CI users

We found reduced activity in the right auditory cortex in CI users relative to controls, which we attribute to differences in auditory stimulation. We limited our sample to CI listeners with unilateral right-sided implants but did not restrict left ear hearing. Most of our subjects with CIs had poor hearing in their left ears, which would result in reduced auditory information being passed to the contralateral (right) auditory cortex. This was as opposed to controls who had bilateral hearing. Prior fNIRS studies have also shown that activity in the superior temporal cortex corresponds with stimulation and comprehension (*Olds et al., 2016*; *Zhou et al., 2018*). We also leave open the possibility that the CI hardware in the right hemisphere interfered with the signal strength, although our simulation studies suggest this cannot fully explain the group differences (Appendix 2 and *Figure 3—figure supplements 1–3*).

What is potentially more interesting is the hint at a lower level of activity in the left auditory cortex of the CI users compared to controls, even though all CI listeners were receiving adequate stimulation of their right auditory nerve with a right CI. There are several possible explanations for this finding. First, activity in superior temporal cortex does not reflect only 'basic' auditory stimulation, but processing related to speech sounds, word meaning, and other levels of linguistic analysis. Thus, although subjects with CIs were certainly receiving stimulation and speech intelligibility scores were generally good, some variability was still present (mean speech perception score in quiet = 0.88, SD = 0.09). The overall level of speech processing was significantly (p=0.00005) lower for CI users than controls (mean speech perception score = 0.99, SD = 0.01), resulting in decreased activity (indeed, because the depth of HD-DOT includes only about 1 cm of the brain, much of primary auditory cortex is not present in our field of view, and the observed group differences were localized in non-primary regions of STG and MTG).

Perhaps the most provocative explanation is that a reduction in top-down modulatory processes (*Davis and Johnsrude, 2007*) plays out as reduced activity in the temporal cortex. That is, given that effortful listening depends on attention (*Wild et al., 2012*), it might be that processes related to top-down prediction (*Cope et al., 2017*; *Sohoglu et al., 2012*; *Sohoglu et al., 2014*) are muted when too much cognitive control is required for perceptual analysis. Reconciling this interpretation with predictive coding accounts of speech perception (*Blank and Davis, 2016*; *Sohoglu and Davis, 2020*) will require additional work. We emphasize that the group differences in left auditory regions were not significant, and thus our interpretation is speculative.

## Individual differences in PFC activation during spoken word recognition

Because of the variability of outcomes in CI users (*Firszt et al., 2004*; *Holden et al., 2013*), one promising thought is that individual differences in brain activation may help explain variability in speech perception ability. Although our study was not powered for individual difference analysis (*Yarkoni and Braver, 2010*), we conducted exploratory correlations to investigate this avenue of inquiry. Interestingly, we saw a trend such that poorer hearing in the left (non-CI) ear was correlated with increased activity in PFC. Our subjects with CIs had significant variability in left ear hearing. Because the speech task was conducted using loudspeakers, we would expect both ears to contribute to accurate perception. Thus, poorer hearing in the left ear would create a greater acoustic challenge, with a correspondingly greater drain on cognitive resources. This interpretation will need additional data to be properly tested.

## Comparison with previous fNIRS studies in CI users

Due to limitations of using fMRI and EEG in studying CI users, fNIRS-based neuroimaging is an attractive method for studying the neural correlates of speech perception in this population (*Hassanpour et al., 2015*; *Lawler et al., 2015*; *Saliba et al., 2016*; *Sevy et al., 2010*; *Zhou et al., 2018*). Some prior studies have looked at relationships of behavioral performance and brain activity in CI users. For example, *Olds et al., 2016*, studied temporal lobe activity and showed that both normal hearing and CI users with good speech perception exhibited greater cortical activations to natural speech than to unintelligible speech. In contrast, CI users with poor speech perception had large, indistinguishable cortical activations to all stimuli. *Zhou et al., 2018*, found that regions of temporal and frontal cortex had significantly different mean activation levels in response to auditory speech in CI listeners compared with normal-hearing listeners, and these activation levels were negatively correlated with CI users' auditory speech understanding.

Our current study differs in that we were able to simultaneously measure responses in temporal and frontal lobes with a high-density array. Our finding of increased recruitment of left PFC in CI listeners is broadly consistent with the recruitment of PFC in fNIRS studies in normal-hearing participants listening to simulated distorted speech (*Defenderfer et al., 2021*; *Lawrence et al., 2018*). Potential differences in anatomical localization could reflect the difference between the type of the distortion created by an actual CI compared to simulated degraded speech listening scenarios in the previous studies, or in source localization accuracy across different imaging hardware. To our knowledge, our study is the first fNIRS-based study to use a non-verbal spatial working memory task for localizing the PFC ROI in the same population, and it may be that increased use of cross-domain functional localizers will prove to be a useful approach in future work.

## Caveats, considerations, and future directions

As with all fNIRS-based functional brain imaging, our ability to image neural activity is limited by the placement of sources and detectors, and in depth to approximately 1 cm of the cortex. Our HD-DOT cap covers the regions of superficial cortex commonly identified in fMRI studies of speech processing, including bilateral superior (STG) and middle temporal gyri (MTG), and pars triangularis of the left IFG (*Davis and Johnsrude, 2003*; *Rodd et al., 2005*; *Rogers et al., 2020*; *Wild et al., 2012*). We also have good coverage of occipital cortex and middle frontal gyrus. However, in the present study we do not image pars opercularis of the IFG, the cerebellum, or any subcortical structures. Thus, it is certainly possible that additional regions not identified here play different roles in understanding speech for CI listeners, a possibility that will require fNIRS setups with improved cortical coverage and converging evidence from other methods to explore.

We also note that some care should be taken in the degree to which we have identified PFC. As we have pointed out, the functional organization of the frontal cortex is complex and differs from person to person; in particular, fMRI studies have demonstrated that regions responding to language and non-language tasks lie nearby each other (*Fedorenko et al., 2012*). Although we attempted to improve the interpretation of our results by using a functional localizer, we were not able to define subject-specific ROIs, which would be preferable. Future work with single-subject localizers and functional ROIs is needed to more clearly resolve this issue.

Cognitive demand during speech understanding frequently comes up in discussions of effortful listening (*Pichora-Fuller et al., 2016*). Although understandable, it is important to keep in mind that the construct of listening effort is not clearly defined (*Strand et al., 2021*) and different measures of 'effort' do not always agree with each other (*Strand et al., 2018*). Here, we interpret increased activity in PFC as reflecting greater cognitive demand. Although we did not include an independent measure of cognitive challenge, we note that simulated CI speech (i.e., noise-vocoded speech) is associated with delayed word recognition (*McMurray et al., 2017*) and poorer memory for what has been heard (*Ward et al., 2016*), consistent with increased dependence on shared cognitive processes (*Piquado et al., 2010*). Relating activity in PFC to cognitive demand is also consistent with *decreased* activity in PFC when speech becomes unintelligible (*Davis and Johnsrude, 2003*).

Finally, we emphasize that in the current study we only measured responses to speech in quiet. Everyday communication frequently occurs in the presence of background noise, which can be particularly challenging for many listeners with CIs (*Firszt et al., 2004*). Exploring how activity in PFC and other regions fluctuates in response to speech at various levels of background noise would be a highly interesting future extension of this work. Based on parametric modulations of acoustic challenge in listeners with normal hearing, we might expect increasing activity in PFC as speech gets more challenging (*Davis and Johnsrude, 2003*), followed by cingulo-opercular activity once intelligibility is significantly hampered (*Vaden et al., 2013*).

**Table 1.** Demographic information.

| Measure | Control | CI users |
|---|---|---|
| Number of subjects (# of females) | 18 (11) | 20 (11) |
| Mean age at test in years (std) | 57.57 (12.74) | 56.80 (14.09) |
| Mean years of CI use (std) | – | 8.10 (6.51) |
| Mean speech perception score (AzBio sentences in quiet) (std) | 0.99 (0.01) | 0.88 (0.09) |
| Mean right ear 4fPTA (std) | 16.02 (6.74) | 21.85 (5.30) with CI on |
| Mean left ear 4fPTA (std) | 16.61 (7.67) | 91.25 (26.77) unaided |
| Mean left ear 4fPTA at test[*,†] (std) | – | 73.28 (37.72) |
| Mean duration of deafness right ear | – | 12.58 (11.74) |

If no response at a given frequency, a value of 120 dB HL was assigned.

[*]With hearing aid, if the subject used amplification. Eight out of 20 CI (cochlear implant) users used hearing aids.

[†]4fPTA (four-frequency pure-tone average), average pure tone threshold at four frequencies (500, 1000, 2000, 4000 Hz).

In summary, using high-density optical neuroimaging, we found increased activity in PFC in listeners with CIs compared to listeners with normal hearing while listening to words in quiet. Our findings are consistent with a greater reliance on domain-general cognitive processing and provide a potential framework for the cognitive effort that many CI users need to expend during speech perception.

## Materials and methods

### Subjects

We recruited 21 adult CI patients aged 26–79 years (17 right-handed, 2 left-handed, 2 not available), and 19 age- and sex-matched controls (18 right-handed, 1 left-handed) (demographic information in *Table 1*). We excluded one CI user due to poor signal quality (evaluated as mean band limited signal-to-noise ratio (SNR) of all source-detectors) and one control due to excessive motion (see Appendix 1 for details). All CI patients had a unilateral right CI (manufacturer: 11 Cochlear, 6 Advanced Bionics, 3 Med-El). All subjects were native speakers of English with no self-reported history of neurological or psychiatric disorders. All aspects of these studies were approved by the Human Research Protection Office (HRPO) of the Washington University School of Medicine. Subjects were recruited from the Washington University campus and the surrounding community (IRB 201101896, IRB 201709126).

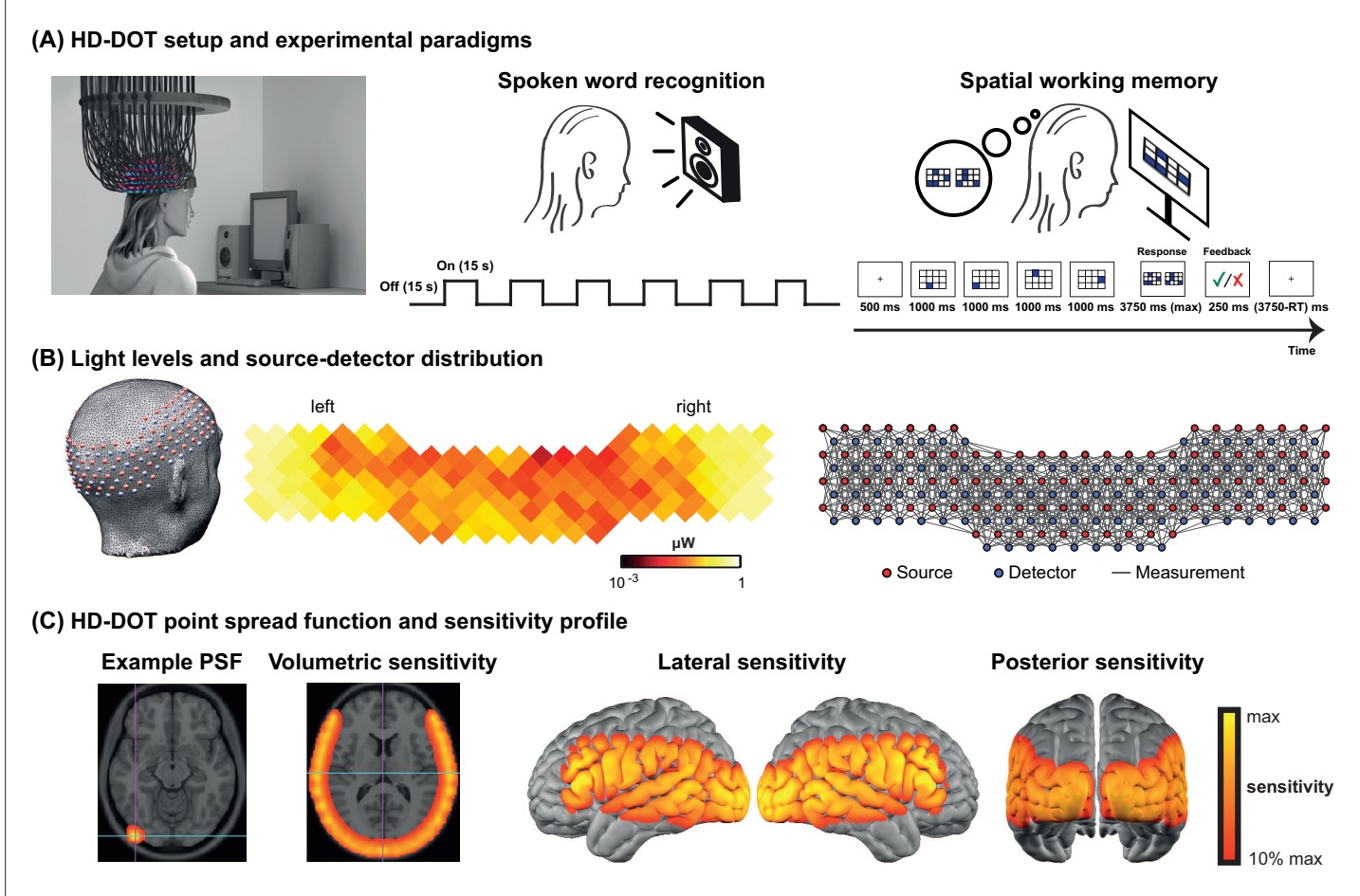

**Figure 5.** High-density diffuse optical tomography (HD-DOT) system and the experimental design. (**A**) Schematic of a subject wearing the HD-DOT cap along with an illustration of the task design. (**B**) Simplified illustration of the HD-DOT system (far left), regional distribution of source-detector light levels (middle), and source-detector pair measurements (~1200 pairs) as gray solid lines illustrated in a flat view of the HD-DOT cap (far right). (**C**) An example point spread function (PSF) and the HD-DOT sensitivity profile illustrated on the volume and spatially registered on the cortical view of a standard atlas in lateral and posterior views.

All subjects gave informed consent and were compensated for their participation in accordance with institutional and national guidelines.

## HD-DOT system

Data were collected using a continuous-wave HD-DOT system comprised of 96 sources (LEDs, at both 750 and 850 nm) and 92 detectors (coupled to APDs, Hamamatsu C5460-01) to enable HbO and HbR spectroscopy (*Figure 5*; *Eggebrecht et al., 2014*). The design of this HD-DOT system provides more than 1200 usable source-detector measurements per wavelength at a 10 Hz full-field frame rate. This system has been validated for successfully mapping cortical responses to language and naturalistic stimuli with comparable sensitivity and specificity to fMRI (*Eggebrecht et al., 2014*; *Fishell et al., 2019*; *Hassanpour et al., 2015*).

## Experimental design

Subjects were seated on a comfortable chair in an acoustically isolated room facing an LCD screen located 76 cm from them, at approximately eye level. The auditory stimuli were presented through two speakers located approximately 150 cm away at about ±21° from the subjects' ears at a sound level of approximately 65 dBA. Subjects were instructed to fixate on a white crosshair against a gray background while listening to the auditory stimuli, holding a keyboard on their lap for the stimuli that required their response (*Figure 5A*, left panel). The HD-DOT cap was fitted to the subject's head to maximize optode-scalp coupling, assessed via real-time coupling coefficient readouts using an in-house software. The stimuli were presented using Psychophysics Toolbox 3 (*Brainard, 1997*) (RRID:SCR_002881) in MATLAB 2010b.

The spoken word recognition paradigm consisted of six blocks of spoken words per run. Each block contained 15 s of spoken words (one word per second), followed by 15 s of silence. Two runs were performed in each study session with a total of 180 words in about 6 min (*Figure 5A*, middle panel). This task was first introduced by *Petersen et al., 1988*, and subsequently replicated in other studies with the same HD-DOT system used in this study (*Eggebrecht et al., 2014*; *Fishell et al., 2019*; *Sherafati et al., 2020*). All sound files were generated from a list of short simple nouns (one to four syllables) pronounced with a female digital-voice audio using AT&T Natural Voices text-to-speech generator with the 'Lauren' digital voice.

To better understand the left PFC activity we observed in our first few CI users, we adopted a spatial working memory task introduced in previous studies (*Fedorenko et al., 2011*; *Fedorenko et al., 2013*) in the remaining subjects to aid in functionally localizing domain-general regions of PFC. In this spatial working memory task, subjects were asked to remember four locations (easy condition) or eight locations (hard condition) in a 3×4 grid, appearing one at a time. Following each trial, subjects had to choose the pattern they saw among two choices, one with correct and one with incorrect locations. This task requires keeping sequences of elements in memory for a brief period and has been shown to activate PFC. Each run for the spatial working memory task was about 8 min, with a total of 48 trials in the run (*Figure 5A*, right panel).

## Data processing

HD-DOT data were pre-processed using the NeuroDOT toolbox (*Eggebrecht and Culver, 2019*). Source-detector pair light level measurements were converted to log-ratio by calculating the temporal mean of a given source-detector pair measurement as the baseline for that measurement. Noisy measurements were empirically defined as those that had greater than 7.5% temporal standard deviation in the least noisy (lowest mean motion) 60 s of each run (*Eggebrecht et al., 2014*; *Sherafati et al., 2020*). The data were next high pass filtered at 0.02 Hz. The global superficial signal was estimated as the average across the first nearest neighbor measurements (13 mm source-detector pair separation) and regressed from all measurement channels (*Gregg et al., 2010*). The optical density time-traces were then low pass filtered with a cutoff frequency of 0.5 Hz to the physiological brain signal band and temporally downsampled from 10 to 1 Hz. A wavelength-dependent forward model of light propagation was computed using an anatomical atlas including the non-uniform tissue structures: scalp, skull, CSF, gray matter, and white matter (*Ferradal et al., 2014*; *Figure 5C*). The sensitivity matrix was inverted to calculate relative changes in absorption at the two wavelengths via reconstruction using Tikhonov regularization and spatially variant regularization (*Eggebrecht et al., 2014*). Relative

changes in the concentrations of oxygenated, deoxygenated, and total hemoglobin (ΔHbO, HbR, ΔHbT) were then obtained from the absorption coefficient changes by the spectral decomposition of the extinction coefficients of oxygenated and deoxygenated hemoglobin at the two wavelengths (750 and 850 nm). After post-processing, we resampled all data to a $3 \times 3 \times 3$ mm$^3$ standard atlas using a linear affine transformation for group analysis. In addition to the standard HD-DOT pre-processing steps used in the NeuroDOT toolbox, we used a comprehensive data quality assessment pipeline (see Appendix 1) to exclude the subjects with low pulse SNR or high motion levels.

After pre-processing, the response for the speech task was estimated using a standard general linear model (GLM) framework. The design matrix was constructed using onsets and durations of each stimulus presentation convolved with a canonical hemodynamic response function (HRF). This HRF was created using a two-gamma function (2 s delay time, 7 s time to peak, and 17 s undershoot) fitted to the HD-DOT data described in a previous study (*Hassanpour et al., 2015*). We included both runs for each subject in one design matrix using custom MATLAB scripts (*Appendix 1—figure 2*).

For modeling the spatial working memory task, we used a design matrix with two columns representing easy and hard conditions. The duration of each easy or hard trial was modeled as the total time of stimulus presentation and evaluation. Events were convolved with the same canonical HRF described in the spoken word perception task to model hemodynamic responses (*Hassanpour et al., 2014*). We used the easy+hard response maps as a reference for defining the PFC ROI, which was more robust than the hard>easy contrast previously used for younger populations.

After estimating the response (β map) for each subject for each task, we performed a second-level analysis in SPM12 (Wellcome Trust Centre for Neuroimaging) version 7487 (RRID:SCR_007037). Extracted time-traces for each subject were then calculated using a finite impulse response model.

We only present the ΔHbO results in the main figures as we have found that the ΔHbO signal exhibits a higher contrast-to-noise ratio compared to ΔHbR or ΔHbT (*Eggebrecht et al., 2014*; *Hassanpour et al., 2014*).

## Functionally defined ROIs

To perform a more focused comparison between controls and CI users, we defined three ROIs, independent from our spoken word recognition dataset, for statistical analysis.

To accurately localize the elevated PFC activation in the CI group, we collected HD-DOT data from nine subjects (four controls and five CI users in 13 sessions) using a spatial working memory task (*Fedorenko et al., 2012*). The visual spatial working memory task robustly activates PFC due to its working memory demands (and visual cortex because of its visual aspect). We chose this task to localize the PFC ROI for performing an ROI-based statistical analysis between controls and CI users. Our logic is that in prior work this task shows activity that dissociates from nearby language-related activity (*Fedorenko et al., 2012*), and thus the region of PFC localized by this task is more likely to reflect domain-general processing (as opposed to language-specific processing). Our results show strong bilateral visual and PFC activations in response to this task (*Figure 6A* left). We then defined the left PFC ROI as the cluster of activation in the left PFC (*Figure 6A* right). This region was centered in inferior frontal sulcus, extending into both dorsal IFG and inferior MFG. The location is broadly consistent with domain-general ('multiple demand') activation (e.g., meta-analysis in *Duncan, 2010*).

To define the left and right auditory ROIs, we used fMRI resting state data from a previously published paper (*Sherafati et al., 2020*) that was masked using the field of view of our HD-DOT system. We defined the left and right auditory ROIs by selecting a 5 mm radius seed in the contralateral hemisphere and finding the Pearson correlation between the time-series of the seed region with all other voxels in the field of view. Correlation maps in individuals were Fisher z-transformed and averaged across subjects (*Figure 6B–C*, left). Right/left auditory ROIs were defined by masking the correlation map to include only the right/left hemisphere (*Figure 6B–C* right). These ROIs extend well beyond primary auditory cortex, which is important to capture responses to words often observed in lateral regions of STG and MTG.

## Speech perception score

To measure auditory-only speech perception accuracy, we presented each participant with two lists of AzBio sentences in quiet (*Spahr et al., 2012*) (except one participant who only heard one set).

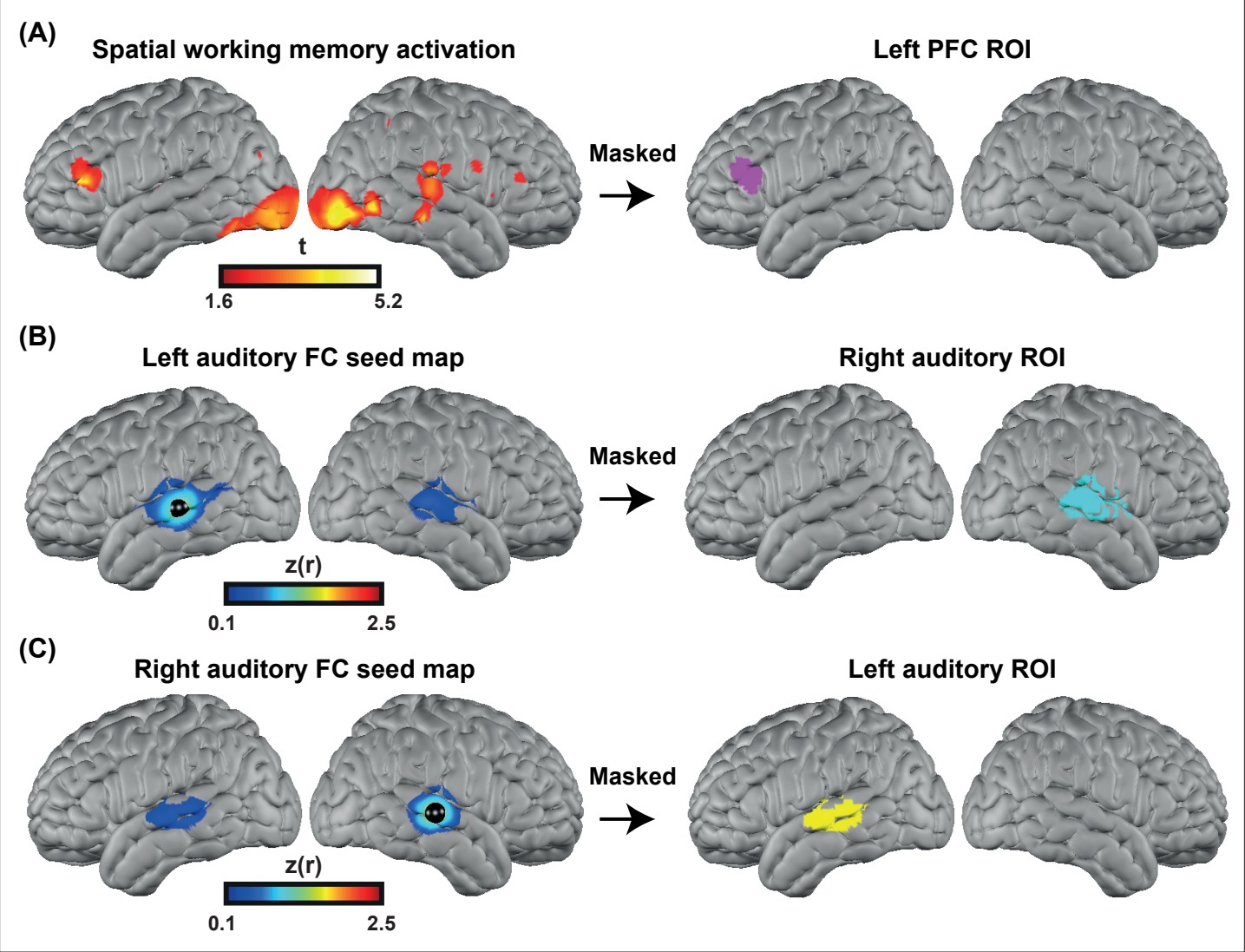

**Figure 6.** Defining functional regions of interest (ROIs). (**A**) Spatial working memory activation for five CI users and four controls over 13 sessions. The prefrontal cortex (PFC) ROI was defined as the cluster of activation in the PFC region, after p<0.05 (uncorrected) voxelwise thresholding. (**B**) Seed-based correlation map for a seed located in the left auditory cortex (left map). Right auditory ROI defined by masking the correlation map to include only the right hemisphere (right map). (**C**) Seed-based correlation map for a seed located in the right auditory cortex (left map). Left auditory ROI defined by masking the correlation map to include only the left hemisphere (right map).

We calculated speech perception accuracy as the proportion of correctly repeated words across all sentences.

### Hearing threshold

We summarized hearing thresholds using a four-frequency pure-tone average (4fPTA), averaging thresholds at 500, 1000, 2000, and 4000 Hz. For unaided testing, a value of 120 dB was assigned if there was no response at a given frequency; for aided or CI testing, a value of 75 dB was used. For CI users, hearing was tested with the right ear CI alone, left ear alone unaided, and left ear with a hearing aid, if worn at the time of testing. *Figure 4—figure supplement 1* shows the audiograms for both controls and CI users.

### Quantification and statistical analysis

Based on prior optical neuroimaging studies with similar speech-related tasks (*Defenderfer et al., 2021*; *Eggebrecht et al., 2014*; *Hassanpour et al., 2015*; *Pollonini et al., 2014*; *Zhou et al., 2018*),

we anticipated 15–20 subjects per group would be sufficient to detect a moderate effect size while also within our resource limitations. We therefore aimed to have at least 15 subjects in each group. We performed a one-tailed two-sample t-test for our main hypothesis (increased recruitment of PFC in CI users), for which we had a strong directional hypothesis (greater activity in listeners with CIs than controls), and a two-tailed t-test for the left and right auditory cortex changes for which we did not have directional hypotheses. We adjusted for unequal variances for left PFC and right auditory ROIs based on the significance of Levene's test.

## Acknowledgements

AS would like to thank Abraham Z Snyder, Andrew K Fishell, Kalyan Tripathy, Karla M Bergonzi, Zachary E Markow, Tracy M Burns-Yocum, Mariel M Schroeder, Monalisa Munsi, Emily Miller, Timothy Holden, and Sarah McConkey for helpful discussions. We also want to thank our participants for their time and interest in our study.

## Additional information

### Competing interests

Jonathan E Peelle: Reviewing editor, eLife. The other authors declare that no competing interests exist.

### Funding

| Funder | Grant reference number | Author |
| --- | --- | --- |
| National Institutes of Health | R21DC015884 | Jonathan E Peelle |
| National Institutes of Health | R21DC016086 | Jonathan E Peelle Joseph P Culver |
| National Institutes of Health | K01MH103594 | Adam T Eggebrecht |
| National Institutes of Health | R21MH109775 | Adam T Eggebrecht |
| National Institutes of Health | R01NS090874 | Joseph P Culver |
| National Institutes of Health | R01NS109487 | Joseph P Culver |
| National Institutes of Health | R01DC019507 | Jonathan E Peelle |

The funders had no role in study design, data collection and interpretation, or the decision to submit the work for publication.

### Author contributions

Arefeh Sherafati, Conceptualization, Data curation, Formal analysis, Methodology, Software, Validation, Visualization, Writing - original draft, Writing – review and editing; Noel Dwyer, Conceptualization, Data curation, Formal analysis, Investigation, Methodology, Project administration, Visualization, Writing – review and editing; Aahana Bajracharya, Investigation, Methodology, Software, Visualization, Writing – review and editing; Mahlega Samira Hassanpour, Conceptualization, Investigation, Software, Writing – review and editing; Adam T Eggebrecht, Conceptualization, Investigation, Methodology, Software, Writing – review and editing; Jill B Firszt, Conceptualization, Formal analysis, Investigation, Methodology, Resources, Supervision, Writing – review and editing; Joseph P Culver, Jonathan E Peelle, Conceptualization, Formal analysis, Funding acquisition, Investigation, Methodology, Resources, Supervision, Writing – review and editing

### Author ORCIDs

Arefeh Sherafati http://orcid.org/0000-0003-2543-0851

Aahana Bajracharya http://orcid.org/0000-0002-7361-6020
Jonathan E Peelle http://orcid.org/0000-0001-9194-854X

### Ethics

Human subjects: All subjects were native speakers of English with no self-reported history of neurological or psychiatric disorders. All aspects of these studies were approved by the Human Research Protection Office (HRPO) of the Washington University School of Medicine. Subjects were recruited from the Washington University campus and the surrounding community (IRB 201101896, IRB 201709126). All subjects gave informed consent and were compensated for their participation in accordance with institutional and national guidelines.

### Decision letter and Author response

Decision letter https://doi.org/10.7554/eLife.75323.sa1
Author response https://doi.org/10.7554/eLife.75323.sa2

## Additional files

### Supplementary files

• Transparent reporting form

### Data availability

Stimuli, data, and analysis scripts are available from https://osf.io/nkb5v/.

The following dataset was generated:

| Author(s) | Year | Dataset title | Dataset URL | Database and Identifier |
|---|---|---|---|---|
| Sherafati A, Bajracharya A, Peelle JE | 2022 | Prefrontal cortex supports speech perception in listeners with cochlear implants | https://osf.io/nkb5v/ | Open Science Framework, https://osf.io/nkb5v/ |

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

# Appendix 1

## Data quality assessment steps

### Motion artifact detection

Motion artifacts were detected in a time-point by time-point manner on the 10 Hz bandpass filtered first nearest neighbor (13 mm) measurements, using a previously described motion detection method in HD-DOT, the global variance of the temporal derivatives (GVTD) (*Sherafati et al., 2018*; *Sherafati et al., 2017*; *Sherafati et al., 2020*).

GVTD, similar to DVARS (derivative of variance) in fMRI (*Power et al., 2012*; *Smyser et al., 2011*), is a vector $g$ that is defined as the RMS of the temporal derivatives $\left(y_{ji} - y_{ji-1}\right)$ across a set of optical density measurements.

$$g = \begin{bmatrix} g_1 \\ \vdots \\ g_M \end{bmatrix}, \; g_i = \sqrt{\frac{1}{N} \sum_{j=1}^{N} \left(y_{ji} - y_{ji-1}\right)^2}, \tag{1}$$

N is the number of first nearest neighbor measurements, and M is the number of time-points. GVTD indexes the global instantaneous change in the time-traces. Higher GVTD values indicate high motion levels. We defined the motion criterion ($g_{thresh}$) (red line in *Appendix 1—figure 1* and 4) based on the GVTD distribution mode ($\widetilde{\kappa}$) plus a constant c = 10 times the standard deviation computed on the left (low) side of the mode ($\sigma_L$).

$$g_{thresh} = \widetilde{\kappa} + c\sigma_L,$$

*Appendix 1—figure 1* shows two example GVTD time-traces for a low-motion and a high-motion run. The GVTD threshold was calculated as $g_{thresh} = \widetilde{\kappa} + 10\sigma_L$ and is shown as a red line.

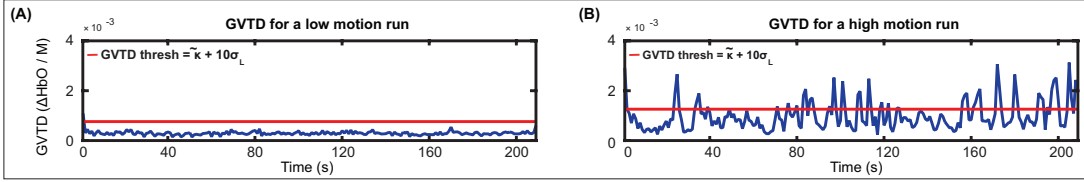

**Appendix 1—figure 1.** Examples of global variance of the temporal derivative (GVTD) time-traces for a low-motion and a high-motion spoken word recognition high-density diffuse optical tomography (HD-DOT) data. The red lines indicate the GVTD threshold of $\widetilde{\kappa} + 10\sigma_L$ of each run.

## Including motion regressors in the design matrix

After determining the time-points that passed the GVTD threshold, we then included a motion regressor as a column for each time-point that passed the motion threshold with one for the noisy time-point, and zeros for every other point in the design matrix, a method commonly known as one-hot encoding. An example design matrix including two runs in a session of spoken word recognition task along with their motion regressors are shown in *Appendix 1—figure 2*.

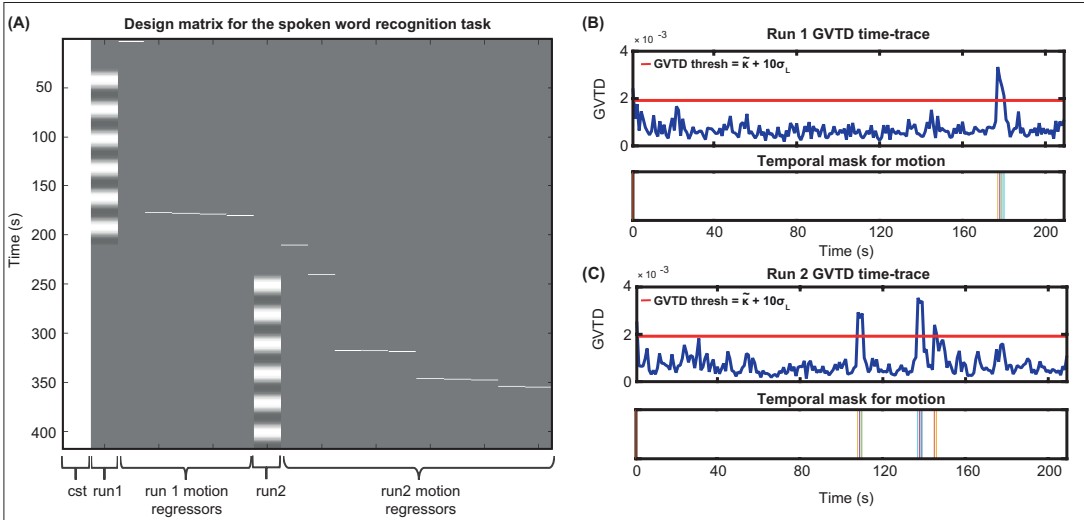

**Appendix 1—figure 2.** Including motion regressors in the design matrix. (**A**) An example design matrix for the general linear model (GLM) for the spoken word recognition task, including a constant column (cst), the task times for run1, one-hot encoding columns for time-points passing the global variance of the temporal derivative (GVTD) threshold for run 1, task times for run2, and one-hot encoding columns for time-points passing the GVTD threshold for run 2. (**B**) The GVTD time-traces and the temporal masks for excluding the time-points passing the GVTD threshold are shown in (**B**) for run 1 and (**C**) for run 2.

## Heartbeat (pulse) SNR across HD-DOT field of view

One indicator of detecting physiological signal in optical neuroimaging is the detection of heartbeat frequency (pulse) (*Sherafati, 2020*). If the data has a very low heartbeat SNR it indicates that the lower frequencies emergent from the hemodynamic fluctuations might also have a low SNR. Therefore, we excluded the subjects that showed a very low mean pulse SNR across the field of view, as it indicates that their brain signal is also less reliable.

The underlying reason behind low pulse SNR is either a poor optode-scalp coupling or high levels of motion. If the optode-scalp coupling is good, normally pulse SNR is only low during high-motion (identified as high GVTD) epochs of data. However, if the optode-scalp coupling is poor, we will see low pulse SNR throughout the run. Note that there are not well-studied definitions and cutoffs for the calculation of pulse SNR in fNIRS-based methods. Therefore, since we already regressed time-points contaminated with motion using the GVTD index, here we only exclude the subjects that had a very low mean pulse SNR across the cap compared to other subjects.

We calculated the pulse SNR using the NeuroDOT function PlotPhysiologyPower.m (https://github.com/WUSTL-ORL/NeuroDOT_Beta; *Eggebrecht and Culver, 2019*). This function first calculates the fast Fourier transform (FFT) of the optical density signal (log mean ratio). Then, it calculates the pulse SNR based on the ratio of the band-limited pulse power ~1 Hz (P) and the noise floor (N).

The band-limited pulse power (P) is defined as the sum of the squares of the FFT magnitudes of a small frequency band around the peak of FFT magnitudes in the 0.5–2 Hz frequency bandwidth. The noise floor (N) is then defined as the median of the squares of the FFT magnitudes of the FFT indices in the same 0.5–1 Hz frequency band, excluding the indices defined as the peak frequency window (used for the heartbeat pulse power). Finally, the pulse SNR was defined as $10 \times \log_{10} (P/N)$.

For pulse SNR calculation, we only used the second nearest neighbor measurements which penetrate through the cortex (30 mm source-detector separation), second wavelength (850 nm), and good measurements (ones with <7.5% std). *Appendix 1—figure 3* shows an example for high SNR values across the field of view (top) and its signature in a subset of the measurements for the same run (bottom) (*Appendix 1—figure 3A*) and an example for low SNR values across the field of view (top) and a subset of measurements for that run (bottom) (*Appendix 1—figure 3B*).

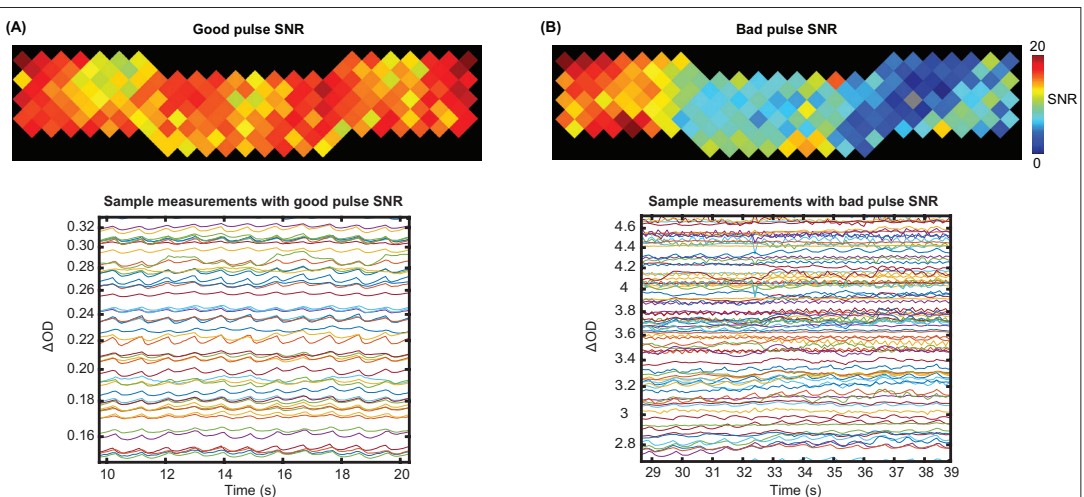

**Appendix 1—figure 3.** Examples of a good and a bad pulse signal-to-noise ratio (SNR) in high-density diffuse optical tomography (HD-DOT) data. Example pulse SNR plot and a selection of the measurements from the HD-DOT array for (**A**) a high-quality pulse SNR, and (**B**) a low-quality pulse SNR. Note the heartbeat frequency (~1 Hz) that appears as around 10 peaks in 10 s.

## Exclusion of subjects based on low pulse and high motion

We rank ordered all subjects based on their mean band limited SNR across the field of view and their mean GVTD values (averaged over two spoken word recognition runs for each subject). We then excluded one of the CI users due to a very low band limited SNR (*Appendix 1—figure 4A*) and one control due to a very high motion level (*Appendix 1—figure 4B*).

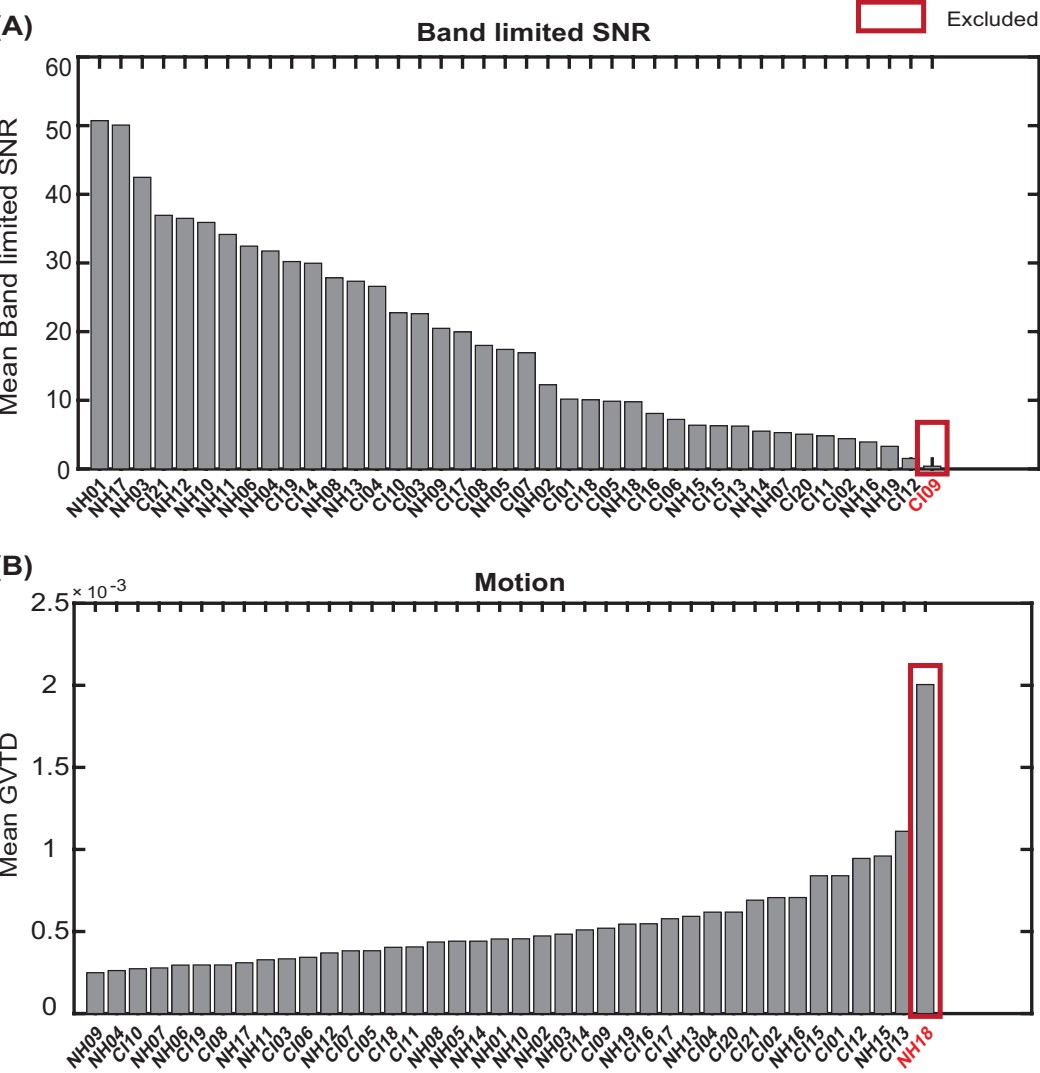

**Appendix 1—figure 4.** Exclusion of subjects with very low band limited signal-to-noise ratio (SNR) and very high motion levels. Sorting all subjects based on (**A**) mean band limited SNR value across the high-density diffuse optical tomography (HD-DOT field of view, and (**B**)) mean global variance of the temporal derivative (GVTD) values across spoken word recognition runs. The red boxes indicate the subjects excluded based on each quality score.

## Appendix 2

### Simulating the effect of the CI transducer blocking the optodes

We identified the approximate area of the HD-DOT field of view that fell under the transducer across people. In most subjects, the transducer only blocked three to five optodes. Due to the high density of the optodes in HD-DOT systems (13 mm nearest source-detector distance), even when an optode is blocked, the light still penetrates through that region from the neighboring sources. Therefore, in most cases, light level plots reveal sensitivity reduction in only two or three optodes. *Appendix 2—figure 1A* shows the location of the CI transducer in one CI user and the optode numbers. *Appendix 2—figure 1B* shows the sensitivity of the HD-DOT cap if all the optodes are fully coupled with the scalp. *Appendix 2—figure 1C and D* demonstrate that the sensitivity drop due to the CI transducer is mainly affecting the IFG and MTG when four and six source-detector pairs are simulated to be completely blocked. Note that the right auditory ROI (*Figure 6B*) mainly includes the STG.

**(A) Example CI user wearing the HD-DOT cap**

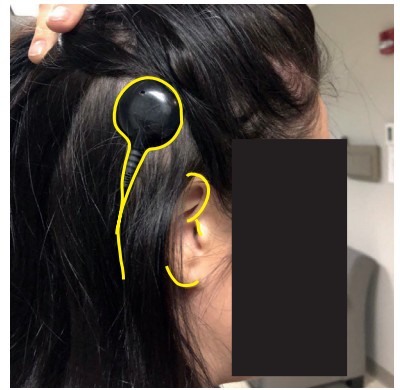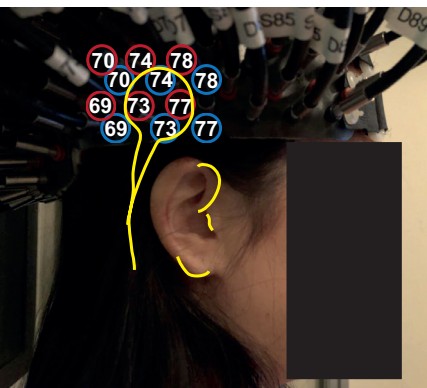

**(B) Full optode array**

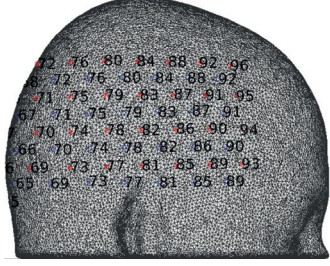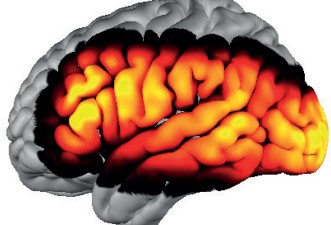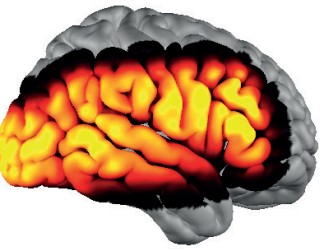

**(C) 4 sources and 4 detectors blocked**

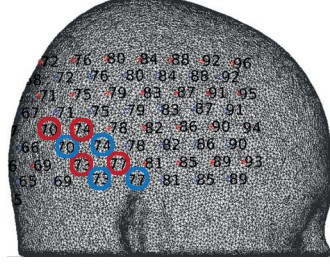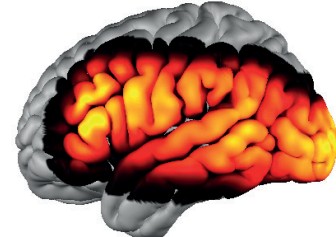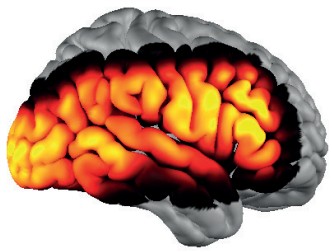

**(D) 6 sources and 6 detectors blocked**

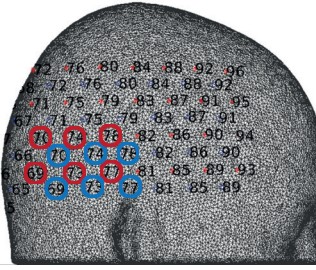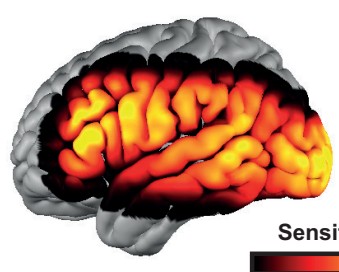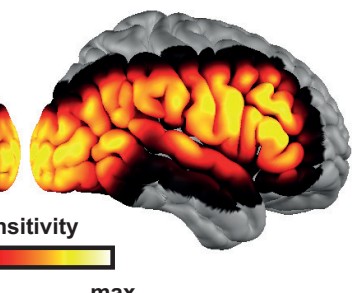

**Sensitivity**

$10^{-3}$ **max**

**Appendix 2—figure 1.** Effects of the cochlear implant (CI) transducer on high-density diffuse optical tomography (HD-DOT) sensitivity. (**A**) A right ear CI user wearing the HD-DOT cap. The yellow contours around the CI transducer with respect to the ear fiducials illustrate the source numbers (encircled in red) and detector numbers (encircled in blue) around the transducer. (**B**) The left panel shows the HD-DOT source-detector grid, overlaid on the mesh used in the study for image reconstruction. The right panel shows the sensitivity of the HD-DOT cap around the cortex including all optodes. (**C**) The left panel shows four sources (in red) and four detectors (in blue) excluded from the sensitivity calculation. The right panel shows the sensitivity of the HD-DOT cap excluding those sources and detectors. (**D**) The left panel shows six sources (in red) and six detectors (in blue) excluded from the sensitivity calculation. The right panel shows the sensitivity of the HD-DOT cap excluding those sources and detectors. Note that the exclusion of sources and detectors in (**C**) and (**D**) only resulted in sensitivity drop off in the inferior and middle temporal gyri.

