## [Editor Report]

The work establishes use of a specific extra area of prefrontal cortex during word listening by CI users and supports a hypothesis based on the multiple demand network that can be tested using other techniques that look at the rest of the network. The revision provides further points of clarity required and better acknowledges the limitations of the technique.

---

## [Decision Letter]

**Decision letter after peer review:**

Thank you for submitting your article "Dorsolateral prefrontal cortex supports speech perception in listeners with cochlear implants" for consideration by *eLife*. Your article has been reviewed by 3 peer reviewers, including Timothy D Griffiths as Reviewing Editor and Reviewer #1, and the evaluation has been overseen by Barbara Shinn-Cunningham as the Senior Editor. The following individual involved in the review of your submission has agreed to reveal their identity: Phillip E Gander (Reviewer #2).

Essential revisions:

The authors should address the following major points in a revision that are highlighted in detail in the individual referee reports.

1. Acknowledgement of and discussion of technical limitations. These are related to the depth of signal and field of view which means that the whole of the network for speech perception is not analysed.

2. Discussion of task effects. There is an effect shown in the frontal cortex listening to the world in quiet but no measurement of correlates of more challenging listening that might show this with the greatest power or demonstrate additional areas. And although listening effort is invoked to explain the data there are no physiological measurements of this.

3. Clearer definition and justification of the ROI within which extra activation occurs. The result in the frontal ROI is just significant without correction for multiple comparisons so it is important that the ROI is clearly justified and explained

*Reviewer #1 (Recommendations for the authors):*

I would like the authors to consider these comments. Basically, I think the demonstration of the DLPFC signal increase during speech perception by the CI group supports models based on the use of additional resources beyond the traditional speech network. An issue for me is that the technique does not allow the examination of other candidate areas.

1. The technique is an advance on fNIRS used previously in terms of spatial resolution but suffers from the same issue with respect to the demonstration of deep sources. This is helpfully shown by the authors in the volumetric field of view diagram in figure 1. The demonstration of the DLPFC involvement in the listening task is robust in the group data including the ROI analysis: this area can be optimally imaged with the technique. The field of view excludes deep parts of the superior temporal plane including the medial parts of the primary auditory cortex and planum temporale behind it, and the deep opercular part of IFG. This prevents the comparison of the signal between CI users and controls in all of the conventional parts of the language network (and other parts of the multiple demand network discussed below).

2. With respect to the decrease in signal in the left auditory cortex in the CI users I think the interpretation is limited because the medial primary cortex is not imaged. I agree with the authors' suggestion that the change here likely reflects a decrease in high-level processing of speech sounds (in the cortex near the convexity) but the discussion might include acknowledgement of the technical limitation due to field of view.

3. The demonstration of increased signal in the left DLPFC is the most robust finding here and is interesting in and of itself. This is consistent with the idea advanced by the senior author that more difficult listening engages anterior cingulate, frontal operculum, premotor cortex, DLPFC and parietal cortex (eg PMID: 28938250 figure 3). Of these areas, only the DLPFC is adequately imaged by this technique (although the authors might comment on whether the inferior parietal cortex is within FOV and demonstrated by contrast in figure 4C). The discussion here focuses more on the multiple-demand network idea and uses a WM task similar to Duncan and others to define this but from this perspective the imaging of the deep operculum, frontal lobe and medial frontal lobe is still missing.

4. I found it hard to interpret the between-subject effect of behavioural scores in figure 6. The effect of left ear hearing (non-CI side) goes in a direction I would predict if the DLPFC signal reflected listening effect and the right ear hearing (CI side) goes in the opposite direction. I guess the range of hearing deficit is broader on the left and this has been emphasised more in the discussion. The authors might comment.

5. Do the authors have any data on varying behaviour within subjects like presenting the sound in noise? Again, any effect on DLPFC would be interesting. But I think the main point here is the use of a different network as a function of CI use even in quiet.

*Reviewer #2 (Recommendations for the authors):*

1. Hypothesis/results interpretation:

Can the justification for the hypothesis be unpacked a little more, for this particular case? If challenging listening requires more processing in a wider array of brain regions to take over for greater cognitive demand, for example in DLPFC. And CI users experience greater cognitive demand during listening, one might hypothesize that listening to single words without background noise would be the hardest condition in which to find an effect (despite the finding of Dwyer et al. 2014). Alternatively, one might compare words in quiet with words in noise, since speech in noise understanding is the main complaint of people with hearing loss (especially CI users). And the brain mechanisms required under these conditions might be different to what is a less challenging task. Perhaps the authors wanted to start in the most "trivial" case and test more conditions later.

Figure 4. The difference t-map has a somewhat loose threshold. Presumably this is because the differences in the frontal cortex would be lost. The authors have previously used more stringent thresholding, but perhaps there is justification here.

The CI>Control t-map shows the strongest differences in the visual/visual association cortex, auditory cortex. On the left it appears that a relevant frontal activation occurs in the inferior frontal gyrus, as might be predicted from the literature.

Reduced activation in the left auditory cortex between CI and control could still be due to threshold differences ~5dB (may be significant). Peripheral fidelity is much lower in the CI group, with also poor contribution ipsilaterally.

The authors propose that lower left auditory cortex activation could also be related to lower speech perception scores, however these scores are drawn from the AzBio test (sentences in noise) which has the potential for indexing very different processes than a word in the quiet test as was the case in this study.

2. ROI definition:

Choosing an ROI functionally defined from a spatial WM task to look for overlap in brain regions sharing a putative cognitive process is unclear if the comparison is a verbal task. If the goal was to find common cognitive resources the referenced paper by Federenko and colleagues achieved such a goal, providing an ROI with which to test against a common set of cognitive processes. Further, the claim that the task activates a brain region, DLFPC, which does seem to include dorsal aspects of inferior frontal gyrus on pars triangularis. Yet the activation found include this region in this study, while this region of pars triangularis does not appear in the work by Federenko and colleagues leave the reader confused about why these differences would exist, unless it was for other reasons, like localization differences between HD-DOT and fMRI.

The results might change considerably if anatomical ROIs were used instead, i.e., IFG or a DLPFC ROI.

It is fine to use the logic that one task (A) primarily indexes cognitive function X (hopefully) and let's see if another task (B) also shares some of that presumed function by masking B with A. That shows (hopefully) that the functions are shared across tasks. In this case, the function of (spatial) working memory is shared between a spatial working memory and speech task in CI users.

DLPFC is an anatomical label, not a functional one. In this case, the authors have used a functional ROI and so need to name it as such, e.g., 'WM', while reference to the term DLPFC is not helpful and should be removed from the manuscript (and title) except perhaps in the discussion.

Line 194. It is slightly confusing to define auditory ROIs from a fMRI resting state paradigm. The referenced paper used a 'word listening' activation for both fMRI and HD-DOT. So are the authors using the acoustic word task rather than rest? And further, if there is data using HD-DOT with a similar paradigm then that might be a more sensible ROI definition?

Line 195-199. The procedure for the definition of the ROI is not clearly motivated, in particular, because the data are obtained from a previous publication which displays thresholded values (at least for the auditory task) that clearly defines the auditory cortex. A correlational approach might (appropriately) be very broad and include areas outside of 'canonical' auditory temporal cortex. Was this intended? What was the threshold for the correlation?

Line 240-245. This text implies the ROI was defined with the current data set rather than another as written in the methods. Please clarify.

Figure 3A. The 'DLPFC' ROI seems to have a large proportion of the cluster within pars triangularis, which is not quite the same as the fMRI overlap representation in Figure 2 of Federnko et al. 2013, in which the bulk of activation is within the middle frontal gyrus.

3. HD-DOT sensitivity:

As stated in the referenced White and Culver paper: "The high-density array has an effective resolution of about 1.3 cm, which should prevent identification to the wrong gyrus."

This means that localization errors to middle frontal gyrus, as opposed to inferior frontal gyrus, could be a concern in this case in which the activation appears to be across a sulcal boundary.

Figure 1. The sensitivity profile appears to show that greater sensitivity exists for the superior aspect of inferior frontal gyrus and inferior aspect of the middle frontal gyrus (and visual cortex). Is that a simple misinterpretation from the figure? Or do the authors know of issues with respect to the HD-DOT technique weighting responses to the regions of greater sensitivity?

Figure 1 supplement. Is it possible there may be some concern about the drop in signal on the right due to variability in the location of the CI transducer across participants? Is it possible the drop in signal in right middle/superior temporal and inferior frontal is also due to loss of measurement here across the group of CI users?

Specific comments:

1. Lines 56-57: Please clarify. Are you saying that listening to degraded input reduces the ability for certain other processing, for instance, performance on a second task that involves memory encoding? Or is it that memory encoding on the speech is limited, or both? It is not entirely clear from the text.

2. What kind of CI device? Since it is not specified, I assume this is a standard length implant and not one for hearing preservation.

3. What was the distribution of CI duration? Is it possible there may be some people under 1 year of use? If so this may be difficult with respect to interpretation for those users, since their performance with the device may not be their best, i.e., they may still yet improve. This process of adaptation to the implant could easily involve different brain regions other than measured in this study, or a different weighting depending on adaptation.

4. Similarly, does the range of unaided hearing in the CI group get to the level of some "functional" residual hearing, or does the skew go to higher thresholds? If there are people with PTAs of ~60 they may perform differently than people without functional residual hearing? Perhaps residual hearing could be added as a covariate to the analyses.

There may also be clues based on performance on the AzBio task. Is there a relationship between residual hearing and AzBio score? The thrust of these questions is that residual hearing likely has an impact on performance, which similarly may influence brain mechanisms.

5. Figure 1. The Spatial WM task appears to have multiple components within it that would be beyond just WM, for example, reward (from the feedback), decision making, response execution, which would all be tied into the level of temporal resolution of the imaging technique. This might temper conclusions about WM per se.

6. Visual activation appears only on the left, which is unclear (compare to Figure 3A). Was the display directly in-front of the participant, or something about their vision?

7. Figure 1. In the spatial working memory task the wait time range of 4000-250ms is unclear.

8. Experimental design. Please provide more information about the following: presentation level for acoustic stimuli, types of words spoken, e.g., monosyllabic, nouns, single talker, gender, etc.

9. Line 149. Confusing text: "…regressed from the all measurement…"

10. Line 176. Was the 'easy+hard' contrast used to provide better SNR?

Alternatively, why not use the ROI defined previously? Did they vary, because if they did that is instructive, perhaps as implied by aging effects?

11. Line 191. Which group of subjects? Is this not the study group as implied on line 176?

12. Line 260. Why does the supplemental analysis not include a contrast of control > CI?

13. Line 304-305. Incomplete sentence.

14. Line 359. The cited literature of similar findings in normal listeners, mostly highlights the inferior frontal gyrus, however, middle frontal gyrus is included in Defenderfer et al.

*Reviewer #3 (Recommendations for the authors):*

Introduction

Line 81: Why predict effects in DLPFC only? There could be a range of 'non-core' brain regions that support speech perception. Were the effects lateralized to left DLPFC predicted in advance of the study?

Methods

Line 91: How were the sample sizes of the CI group and the control group determined? I note that page 7 explains that 15-20 subjects per group was thought to be sufficient, based on similar studies. Did these previous studies include a cross-sectional design?

Page 4. Table 1 reports left ear 4fPTA (unaided). The footnote for Table 1 reports that 8/20 CI users had a hearing aid but did they use them during the speech task?

Line 120. Where were the 2 speakers located relative to the participants? +/- 45 degrees? Which sound level were the words presented at?

Lines 127-130. More details about the spoken word task is needed. Is this a standardized test or a novel test that the authors developed themselves?

Line 131. Which 'preliminary results'?

Line 191. "…based on the response … in a group of subjects…". Which subjects? Merge with relevant information on page 8.

Line 216. Is "Pollonini et al. 2014" cited correctly?

Line 238. Why define only left DLPFC as an ROI?

Page 9. The visual spatial working memory task also activated 'auditory' regions in the right hemisphere that look like they overlap with the 'right auditory ROI' created by the authors. Do the authors consider this overlap to be problematic?

Page 9. Legend for Figure 3. "…in the DLPFC region, survived after…". A word is missing.

Page 10. Figure 4C suggests that CI > controls contrast identified quite a few group differences. So why then focus on left DLPFC?

Page 11. Results of two-sample t-tests are reported in the legend of Figure 5. This information should be taken out of the legend and reported in full. Why were t-tests more appropriate than an ANOVA? Please report results of statistical analyses fully i.e. not just p-values. Were reported p-vals corrected for the number of t-tests that were used?

Page 11. The authors report the results of correlation analyses. Please add further details e.g. are Pearson's correlation coefficients reported?

Page 11. The authors report that the threshold used for statistical significance in the correlation analyses was 'uncorrected'. Presumably this means that reported p-vals were not corrected for the 4 correlation analyses shown?

Page 12. Correlation analyses for CI users. The rationale for these analyses is missing. Why carry out simple correlation analyses on data for CI users only? Potential predictors of speech processing (4PTA) could have been used as a covariate in the main analyses (e.g. Figure 5).

---

## [Author Response]

Essential revisions:The authors should address the following major points in a revision that are highlighted in detail in the individual referee reports.1. Acknowledgement of and discussion of technical limitations. These are related to the depth of signal and field of view which means that the whole of the network for speech perception is not analysed.

Thank you for encouraging us to clarify the limitations in our approach. Of course, we completely agree that though we have good coverage relative to other optical neuroimaging approaches, we do not cover the full brain, and there are indeed brain regions we might expect to be involved in speech perception that we do not image. We have added the following to the Discussion:

“As with all fNIRS-based functional neuroimaging, our ability to image neural activity is limited by the placement of sources and detectors, and in depth to approximately 1 cm of the cortex. Our HD-DOT cap covers the regions of superficial cortex commonly identified in fMRI studies of speech processing, including bilateral superior and middle temporal gyri, and pars triangularis of the left inferior frontal gyrus (Davis and Johnsrude 2003; Rodd, Davis, and Johnsrude 2005; Rogers et al. 2020; Wild et al. 2012). We also have good coverage of occipital cortex and middle frontal gyrus. However, in the present study we do not image pars opercularis of the inferior frontal gyrus, the cerebellum, or any subcortical structures. Thus, it is certainly possible that additional regions not identified here play different roles in understanding speech for CI listeners, a possibility that will require fNIRS setups with improved cortical coverage and converging evidence from other methods to explore.”

2. Discussion of task effects. There is an effect shown in the frontal cortex listening to the world in quiet but no measurement of correlates of more challenging listening that might show this with the greatest power or demonstrate additional areas. And although listening effort is invoked to explain the data there are no physiological measurements of this.

Thank you for raising these two important points.

Regarding the first point, it is important to note that both CI simulations (i.e., noise vocoding) and data from CI listeners show difficulty understanding speech in quiet (Willis, Helen; 2018; The feasibility of the dual-task paradigm as a framework for a clinical test of listening effort in cochlear implant users. Doctoral thesis, University College London). Thus, our perspective is that speech may be “in quiet” in the room, but to the auditory system of a listener with a cochlear implant, the acoustic signal is degraded, and it is precisely the brain’s response to that degraded signal we are measuring. We readily agree that further exploring responses to speech in noise (which, of course, is common in everyday listening, and can be particularly challenging for many listeners with cochlear implants) is highly interesting, and we hope a future extension of this work. We now explicitly note these points in the introduction (to clarify the logic behind studying speech in quiet) and in the Discussion:

“Finally, we emphasize that we report here only responses to speech in quiet. Everyday communication frequently occurs in the presence of background noise, which can be particularly challenging for many listeners with cochlear implants (Firszt et al. 2004). Exploring how activity in PFC and other regions fluctuates in response to speech at various levels of background noise would be a highly interesting future extension of this work. Based on parametric modulations of acoustic challenge in listeners with normal hearing, we might expect increasing activity in PFC as speech gets more challenging (Davis and Johnsrude 2003), followed by cingulo-opercular activity once intelligibility is significantly hampered (Vaden et al. 2013).”

With respect to the second point, it is entirely correct that we do not have a non-imaging measure of cognitive demand during listening. That being said, we think there are good reasons to think that differences here are driven by a response to acoustic challenge, which we believe falls comfortably under the “listening effort” umbrella. However, there are two important clarifications we should make. First, we readily acknowledge that listening effort is not clearly defined (Strand et al., 2020), and in its fullest definition encompasses more than just cognitive challenge (Pichora-Fuller et al., 2016). In fact, in the manuscript we never use the term “listening effort” to describe our paradigm for precisely this reason. We have also done our best to explain where our cognitive-demand-centered interpretation comes from. In short, from imaging studies showing parametric responses to acoustic challenge (including noise vocoding) (e.g., Davis and Johnsrude, 2003), behavioral studies showing that acoustic challenge hinders memory for words (Rabbitt, 1968) and stories (Ward et al., 2016); and the many reports of listeners with CIs having difficulty understanding speech (Firszt et al., 2004, McMurray et al. 2017, and many others). We have added these points to the Discussion:

“Cognitive demand during speech understanding frequently comes up in discussions of effortful listening (Pichora-Fuller et al. 2016). Although understandable, it is important to keep in mind that the construct of listening effort is not clearly defined (Strand et al. 2020) and different measures of “effort” do not always agree with each other (Strand et al. 2018). Here, we interpret increased activity in PFC as reflecting greater cognitive demand. Although we did not include an independent measure of cognitive challenge, we note that simulated CI speech (i.e., noise-vocoded speech) is associated with delayed word recognition (McMurray, Farris-Trimble, and Rigler 2017) and poorer memory for what has been heard (Ward et al., 2016), consistent with increased dependence on shared cognitive processes (Piquado et al. 2010). Relating activity in PFC to cognitive demand is also consistent with decreased activity in PFC when speech becomes unintelligible (Davis and Johnsrude 2003).”

3. Clearer definition and justification of the ROI within which extra activation occurs. The result in the frontal ROI is just significant without correction for multiple comparisons so it is important that the ROI is clearly justified and explained

Thank you for giving us the opportunity to clarify these important points. We start by discussing the definition and justification for the PFC region of interest, followed by a description of a revised analysis.

We have added some additional detail to the justification of the prefrontal cortex ROI. The full revised text reads:

“To accurately localize the elevated prefrontal cortex activation in the CI group, we collected HD-DOT data from 9 subjects (4 controls and 5 CI users in 13 sessions) using a spatial working memory task (Fedorenko, Duncan, and Kanwisher 2012). The visual spatial working memory task robustly activates PFC due to its working memory demands (and visual cortex because of its visual aspect). We chose this task to localize the PFC ROI for performing an ROI-based statistical analysis between controls and CI users. Our logic is that in prior work this task shows activity that dissociates from that in inferior frontal gyrus (Fedorenko, Duncan, and Kanwisher 2012), and thus the region of PFC localized by this task is more likely to reflect domain-general processing (as opposed to language-specific processing). Our results show strong bilateral visual and PFC activations in response to this task (Figure 6A left). We then defined the left PFC ROI as the cluster of activation in the left PFC region (Figure 6A right). This region was centered in inferior frontal sulcus, extending into both dorsal IFG and inferior MFG. The location is broadly consistent with domain-general (“multiple demand”) activation (e.g., meta-analysis in Duncan, 2010).”

In terms of anatomy, our frontal cortex ROI is centered in the inferior frontal sulcus (IFS), which also corresponds to peak activity in meta analyses of “multiple demand” tasks—for example, in this meta-analysis from Duncan (2010, Trends in Cognitive Science).

**Author response image 1. sa2fig1:** 

Being centered in IFS, there is some overlap with both IFG and MFG. If we trust our source localization, we don’t think the IFG overlap is inferior enough to suggest language-related processing, although we concede we do not have definitive evidence for this. The ROI is unquestionably in the prefrontal cortex. In our view, centered in IFS is sufficient for considering this dorsolateral prefrontal cortex. However, we have gone with the more general “prefrontal cortex” in the title and text of the manuscript because we don’t think this point affects our main findings or interpretation.Regarding correction for multiple comparisons, we have revisited this section and clarified that the critical comparison in PFC is corrected for the ROIs looked at.

Reviewer #1 (Recommendations for the authors):I would like the authors to consider these comments. Basically, I think the demonstration of the DLPFC signal increase during speech perception by the CI group supports models based on the use of additional resources beyond the traditional speech network. An issue for me is that the technique does not allow the examination of other candidate areas.1. The technique is an advance on fNIRS used previously in terms of spatial resolution but suffers from the same issue with respect to the demonstration of deep sources. This is helpfully shown by the authors in the volumetric field of view diagram in figure 1. The demonstration of the DLPFC involvement in the listening task is robust in the group data including the ROI analysis: this area can be optimally imaged with the technique. The field of view excludes deep parts of the superior temporal plane including the medial parts of the primary auditory cortex and planum temporale behind it, and the deep opercular part of IFG. This prevents the comparison of the signal between CI users and controls in all of the conventional parts of the language network (and other parts of the multiple demand network discussed below).

We acknowledge the limitation of our system field of view in covering the entirety of brain networks involved in speech processing, particularly deep structures. We agree this is an important point, and now note this explicitly in the Discussion.

2. With respect to the decrease in signal in the left auditory cortex in the CI users I think the interpretation is limited because the medial primary cortex is not imaged. I agree with the authors' suggestion that the change here likely reflects a decrease in high-level processing of speech sounds (in the cortex near the convexity) but the discussion might include acknowledgement of the technical limitation due to field of view.

This comment is very important and we thank the reviewer for reminding us of this fact. We have added a sentence in the Discussion section to address this point.

3. The demonstration of increased signal in the left DLPFC is the most robust finding here and is interesting in and of itself. This is consistent with the idea advanced by the senior author that more difficult listening engages anterior cingulate, frontal operculum, premotor cortex, DLPFC and parietal cortex (eg PMID: 28938250 figure 3). Of these areas, only the DLPFC is adequately imaged by this technique (although the authors might comment on whether the inferior parietal cortex is within FOV and demonstrated by contrast in figure 4C). The discussion here focuses more on the multiple-demand network idea and uses a WM task similar to Duncan and others to define this but from this perspective the imaging of the deep operculum, frontal lobe and medial frontal lobe is still missing.

We focused on the PFC ROI for two reasons. First, our reading of prior fMRI literature (and work from our own group) suggests that when speech is challenging to understand, but still intelligible, increases are seen in many regions of the dorsolateral prefrontal cortex. However, when intelligibility begins to suffer (i.e., errors are introduced), the cingulo-opercular network is engaged (see compelling fMRI work from Eckert, Vaden, and colleagues). Because we were presenting speech in quiet, to CI users with fairly good auditory-only perception, we focused on the PFC activity. As you point out, we also do not have the ability to image deep areas—and so we are not particularly well-suited for studying the cingulo-opercular network.

We had considered including an inferior parietal ROI, which as you say, is also often implicated in multiple-demand tasks. However, we did not have a clear ROI defined based on our spatial working memory task. Thus, we did not feel justified in including one. (It is also unclear whether our parietal coverage adequately covers IPS—it may, but the absence of strong functional activity here led us to abandon this idea for now.)

4. I found it hard to interpret the between-subject effect of behavioural scores in figure 6. The effect of left ear hearing (non-CI side) goes in a direction I would predict if the DLPFC signal reflected listening effect and the right-ear hearing (CI side) goes in the opposite direction. I guess the range of hearing deficit is broader on the left and this has been emphasised more in the discussion. The authors might comment.

We agree with this comment. We also think that our rather small sample size and the small variability in the right ear hearing levels, makes our PFC correlation with right ear 4fPTA harder to interpret. We added a sentence to state this more clearly in the discussion.

5. Do the authors have any data on varying behaviour within subjects like presenting the sound in noise? Again, any effect on DLPFC would be interesting. But I think the main point here is the use of a different network as a function of CI use even in quiet.

Because we have brain imaging data for speech in quiet, we felt that the strongest justification for brain-behavior correlations would be to use behavioral scores from a similar condition.

Reviewer #2 (Recommendations for the authors):1. Hypothesis/results interpretation:Can the justification for the hypothesis be unpacked a little more, for this particular case? If challenging listening requires more processing in a wider array of brain regions to take over for greater cognitive demand, for example in DLPFC. And CI users experience greater cognitive demand during listening, one might hypothesize that listening to single words without background noise would be the hardest condition in which to find an effect (despite the finding of Dwyer et al. 2014). Alternatively, one might compare words in quiet with words in noise, since speech in noise understanding is the main complaint of people with hearing loss (especially CI users). And the brain mechanisms required under these conditions might be different to what is a less challenging task. Perhaps the authors wanted to start in the most "trivial" case and test more conditions later.

As we note above, speech in quiet is still degraded from the perspective of the auditory system of a CI user. The brain responses to degraded speech are not uniform, but depend (among other things) on the overall challenge and intelligibility. We focused on speech in quiet to (a) demonstrate differences in processing for listeners with CIs “even” in quiet, and (b) to maximize intelligibility, which in our view helps interpretation (i.e., we answer the question: what are listeners doing to understand speech when they are being successful?).

Figure 4. The difference t-map has a somewhat loose threshold. Presumably this is because the differences in the frontal cortex would be lost. The authors have previously used more stringent thresholding, but perhaps there is justification here.

Because we had rather specific anatomically-constrained hypotheses, we constructed the study first and foremost around an ROI-based analysis. However, we also thought it was important to show non-ROI results. Hence, in Figure 2 we include unthresholded maps of parameter estimates, and thresholded (but not corrected) t maps. We do this to give a sense of the extent of the data and to be transparent about what is, and isn’t showing up at the whole-brain level. It is correct that the result in PFC does not survive correcting for multiple comparisons at the whole brain level.

The CI>Control t-map shows the strongest differences in the visual/visual association cortex, auditory cortex. On the left it appears that a relevant frontal activation occurs in the inferior frontal gyrus, as might be predicted from the literature.Reduced activation in the left auditory cortex between CI and control could still be due to threshold differences ~5dB (may be significant). Peripheral fidelity is much lower in the CI group, with also poor contribution ipsilaterally.

An additional reason we provide the whole-brain maps is to help readers think about additional hypotheses and guide future research. We agree that activity in IFG is also of interest.

The authors propose that lower left auditory cortex activation could also be related to lower speech perception scores, however these scores are drawn from the AzBio test (sentences in noise) which has the potential for indexing very different processes than a word in the quiet test as was the case in this study.

We used the AzBio sentences, but these were presented in quiet to get a measure of speech-in-quiet understanding. We have clarified this in the revised manuscript.

2. ROI definition:Choosing an ROI functionally defined from a spatial WM task to look for overlap in brain regions sharing a putative cognitive process is unclear if the comparison is a verbal task. If the goal was to find common cognitive resources the referenced paper by Federenko and colleagues achieved such a goal, providing an ROI with which to test against a common set of cognitive processes. Further, the claim that the task activates a brain region, DLFPC, which does seem to include dorsal aspects of inferior frontal gyrus on pars triangularis. Yet the activation found include this region in this study, while this region of pars triangularis does not appear in the work by Federenko and colleagues leave the reader confused about why these differences would exist, unless it was for other reasons, like localization differences between HD-DOT and fMRI.The results might change considerably if anatomical ROIs were used instead, i.e., IFG or a DLPFC ROI.It is fine to use the logic that one task (A) primarily indexes cognitive function X (hopefully) and let's see if another task (B) also shares some of that presumed function by masking B with A. That shows (hopefully) that the functions are shared across tasks. In this case, the function of (spatial) working memory is shared between a spatial working memory and speech task in CI users.DLPFC is an anatomical label, not a functional one. In this case, the authors have used a functional ROI and so need to name it as such, e.g., 'WM', while reference to the term DLPFC is not helpful and should be removed from the manuscript (and title) except perhaps in the discussion.

We have updated the term to “prefrontal cortex” because this region is unambiguously in the prefrontal cortex. In our view, it is in the dorsal and lateral aspect of the prefrontal cortex, which motivated our use of DLPFC. However we acknowledge there is no clear consensus on DLPFC boundaries and we don’t find this label critical to our argument.

Line 194. It is slightly confusing to define auditory ROIs from a fMRI resting state paradigm. The referenced paper used a 'word listening' activation for both fMRI and HD-DOT. So are the authors using the acoustic word task rather than rest? And further, if there is data using HD-DOT with a similar paradigm then that might be a more sensible ROI definition?

We appreciate the reviewer's attention to this point. In the referenced paper, both word listening and resting state data were presented. We made Author response image 2 to show that there is a large overlap between the left and right auditory ROIs found using the seed based resting state maps and the auditory maps in response to the hearing words task. The dice coefficient between our proposed resting state based ROI and the hearing words based ROI (thresholded at p = 0.05) is 0.7 and the voxelwise similarity of the two is also 0.7. We chose the resting state based ROIs based on fMRI data, because it is independent of any particular task, although we see the alternative logic of using the same task, as well. In practice regions defined both ways are similar, and do not affect our conclusions.

Line 195-199. The procedure for the definition of the ROI is not clearly motivated, in particular, because the data are obtained from a previous publication which displays thresholded values (at least for the auditory task) that clearly defines the auditory cortex. A correlational approach might (appropriately) be very broad and include areas outside of 'canonical' auditory temporal cortex. Was this intended? What was the threshold for the correlation?

It is not uncommon for speech-related activity to extend beyond the superior temporal plane, through STG and into STS (and sometimes MTG). Our aim was to define an ROI that covered this span of auditory responses and corresponded to our prior work using this particular DOT system (see, e.g., Eggebrecht et al. 2014). We found the ROIs defined this way showed good general correspondence to the activation in our speech tasks, which was our specific use case. For thresholding the ROIs we used a threshold of z(r) = 0.1.

Line 240-245. This text implies the ROI was defined with the current data set rather than another as written in the methods. Please clarify.

Thanks for bringing this ambiguity to our attention. To make this point clear, we consolidated “Region of interest analysis” in Methods and “Functionally defined ROIs” in Results to ““Functionally defined regions of interest”” in Methods.

There, we clarified that:

“To accurately localize the elevated PFC activation in the CI group, we collected HD-DOT data from 9 subjects (4 controls and 5 CI users in 13 sessions) using a spatial working memory task (Fedorenko et al., 2012)” …. “To define the left and right auditory ROIs, we used a previously published fMRI resting state dataset (Sherafati et al., 2020)”

Figure 3A. The 'DLPFC' ROI seems to have a large proportion of the cluster within pars triangularis, which is not quite the same as the fMRI overlap representation in Figure 2 of Federnko et al. 2013, in which the bulk of activation is within the middle frontal gyrus.

One strength of using a functional localizer is that our inferences are most strongly driven by the overlap in activity, rather than putative anatomy. Though, we also note that our ROI is centered on IFS, which is often the center of overlap in multiple-demand network studies. However, there certainly could be some anatomical overlap with portions of the IFG.

3. HD-DOT sensitivity:As stated in the referenced White and Culver paper: "The high-density array has an effective resolution of about 1.3 cm, which should prevent identification to the wrong gyrus."This means that localization errors to middle frontal gyrus, as opposed to inferior frontal gyrus, could be a concern in this case in which the activation appears to be across a sulcal boundary.

Our interpretation of the prefrontal activity in CI users is not based primarily on the anatomical landmark, but by the functional definition of the region (which we interpret in an anatomical context).

Figure 1. The sensitivity profile appears to show that greater sensitivity exists for the superior aspect of inferior frontal gyrus and inferior aspect of the middle frontal gyrus (and visual cortex). Is that a simple misinterpretation from the figure? Or do the authors know of issues with respect to the HD-DOT technique weighting responses to the regions of greater sensitivity?

The variation in sensitivity profile proceeds from the arrangement of the sources and detectors. Near the edge of the field of view, sensitivity begins to drop. With our current cap configuration, this does result in small sensitivity differences (Appendix 2– figure 1B). However, we apply a more stringent field of view mask which was created using the intersection of sensitivity across multiple participants to all our data to exclude any voxels with a sensitivity less than around 10% of the maximum sensitivity (Figure 5C). This avoids including any voxels that might be less reliable due to inhomogeneous sensitivity.

Figure 1 supplement. Is it possible there may be some concern about the drop in signal on the right due to variability in the location of the CI transducer across participants? Is it possible the drop in signal in right middle/superior temporal and inferior frontal is also due to loss of measurement here across the group of CI users?

Thank you for raising this important point, which is something we’ve worried about a lot. Although we cannot rule out the effects of lower sensitivity in the magnitude of the activation in certain areas, we were fairly confident that the ROIs we explored are not overlapping with low sensitivity areas.

To quantify the expected change in signal resulting from the CI hardware, in addition to our simulation of two levels of optode blocking (moderate case 4 sources and 4 detectors, and extreme case 6 sources and 6 detectors, now in Appendix 2 – figure 1), we added another simulation study (Figure 3—figure supplements 1-3). Briefly, for each CI user, we calculated how many sources and detectors were affected by the hardware. We then excluded this number of sources/detectors from a control participant. We repeated this process 100 times with different random assignments of excluded channels to the control data to get a sense of what the time course would be for control data (but missing the same number of channels as our CI data). We found a small drop in auditory activation that was still significantly above what we observed in the CI listeners. We note that we cannot completely rule out the effects of CI hardware on signals in the right temporal and parietal regions; however, we do not think it can fully account for our findings in the right auditory cortex. (Because we only studied CI users with a right unilateral implant, regions in the left hemisphere are unaffected.)

Specific comments:1. Lines 56-57: Please clarify. Are you saying that listening to degraded input reduces the ability for certain other processing, for instance, performance on a second task that involves memory encoding? Or is it that memory encoding on the speech is limited, or both? It is not entirely clear from the text.

When speech is degraded, listeners have trouble remembering it. For example, in Ward et al. (2016) we played normal-hearing participants short stories that were either normal speech, or degraded (but intelligible) due to noise vocoding (16 or 20 channels). We asked participants to repeat back as much of the stories as they could, and found poorer memory for the degraded stories (despite an intelligibility test indicating the stories were intelligible). Perhaps even more compelling evidence comes from studies of word list memory, in which not only the degraded word, but *subsequent* words are remembered more poorly. Because the subsequent words are not degraded, it is difficult to come up with an acoustic explanation for the memory difficulty—thus pointing towards some shared cognitive resource needed to understand degraded speech, that is thus less available for memory encoding.

2. What kind of CI device? Since it is not specified, I assume this is a standard length implant and not one for hearing preservation.

11 used Cochlear, 6 used Advanced Bionics, 3 used Med EI. We have added this information to the methods.

3. What was the distribution of CI duration? Is it possible there may be some people under 1 year of use? If so this may be difficult with respect to interpretation for those users, since their performance with the device may not be their best, i.e., they may still yet improve. This process of adaptation to the implant could easily involve different brain regions other than measured in this study, or a different weighting depending on adaptation.

We agreed the duration of use is important to consider. We only included CI users if they had at least 1 year of experience with their implant. The years of experience ranged between 1-27.

**Author response image 3. sa2fig3:** 

We performed some exploratory analyses using this but did not see anything compelling. In general we don’t think this study is adequately powered for individual difference analysis, but experience is certainly a variable we are keeping squarely in mind for future studies.

4. Similarly, does the range of unaided hearing in the CI group get to the level of some "functional" residual hearing, or does the skew go to higher thresholds? If there are people with PTAs of ~60 they may perform differently than people without functional residual hearing? Perhaps residual hearing could be added as a covariate to the analyses.There may also be clues based on performance on the AzBio task. Is there a relationship between residual hearing and AzBio score? The thrust of these questions is that residual hearing likely has an impact on performance, which similarly may influence brain mechanisms.

We agree regarding the importance of residual hearing and its variability.

We found no significant correlation between the speech perception score (average AzBio score in quiet) and the residual hearing, either defined as left ear 4fPTA unaided, or left ear 4fPTA (aided).

**Author response image 4. sa2fig4:** 

5. Figure 1. The Spatial WM task appears to have multiple components within it that would be beyond just WM, for example, reward (from the feedback), decision making, response execution, which would all be tied into the level of temporal resolution of the imaging technique. This might temper conclusions about WM per se.

We agree the spatial working memory task encompasses multiple processes. In interpreting this task, and the areas involved, we are relying on the fairly extensive multiple-demand literature tied to these regions, and the work of Fedorenko and colleagues showing that—in single-subject applications—regions activated by this task dissociate from those activated by language tasks. But, also, we have tried to remain somewhat agnostic about the specific cognitive processes engaged. We have made some educated guesses based on the broader literature, but we agree that our own current data have to be understood in terms of what we actually tested.

6. Visual activation appears only on the left, which is unclear (compare to Figure 3A). Was the display directly in-front of the participant, or something about their vision?

The display was directly in front of the participant. The crosshair at the center of the monitor has been aligned to the middle point between the eyes of the participant. We do see a stronger visual activation on the left side, however, we did not collect any vision data and just have verbal confirmation of the participant’s normal vision.

However, Author response image 5 is the same spatial working memory activation (same map depicted in Figure 1C) with a lower t threshold (p = 0.05, t=1.6) showing some activation on the right side which did not survive the more stringent thresholding. Thus we are reluctant to make too much of this qualitative asymmetry.

**Author response image 5. sa2fig5:** 

7. Figure 1. In the spatial working memory task the wait time range of 4000-250ms is unclear.

We apologize for the confusion. The time is 3750 ms (i.e., 4000 – 250), further subtracting the response time. In this way, the “response + feedback + fixation” portion of each trial is kept to 4000 ms on each trial. We modified the figure text to clarify this.

8. Experimental design. Please provide more information about the following: presentation level for acoustic stimuli, types of words spoken, e.g., monosyllabic, nouns, single talker, gender, etc.

Now added to the revised manuscript.

9. Line 149. Confusing text: "…regressed from the all measurement…"

Thank you for catching this, now repaired.

10. Line 176. Was the 'easy+hard' contrast used to provide better SNR?

That is correct.

Alternatively, why not use the ROI defined previously? Did they vary, because if they did that is instructive, perhaps as implied by aging effects?

If you mean the ROI defined in past studies using fMRI, we had three reasons for using our own ROI. First, we are moving towards subject-specific localizers, and as such, need to collect these data in all of our own participants. Second, our participants indeed differ in age from most prior reports with this task, and this may affect the extent of activation. Third, given the intricate anatomy of the prefrontal cortex we wanted to use a within-modality definition for this ROI.

11. Line 191. Which group of subjects? Is this not the study group as implied on line 176?

As mentioned in lines 359-362, we did not plan on collecting data for the spatial working memory task in the beginning of this study. “To better understand the left PFC activity we observed in our first few CI users, we adopted a spatial working memory task introduced in previous studies (Fedorenko et al., 2011; Fedorenko et al., 2013) in the remaining subjects to aid in functionally localizing domain-general regions of prefrontal cortex.” Since in the literature the variability across individuals has been reported for this task, we aimed at defining an ROI based on at least a subset of our own demographic cohort of interest. Second, we wanted to use the same imaging modality to localize this activation.

12. Line 260. Why does the supplemental analysis not include a contrast of control > CI?

Added.

13. Line 304-305. Incomplete sentence.

Thank you for catching this, now fixed.

14. Line 359. The cited literature of similar findings in normal listeners, mostly highlights the inferior frontal gyrus, however, middle frontal gyrus is included in Defenderfer et al.

We categorize this activity as “prefrontal cortex” to cover both IFG and MFG here.

Reviewer #3 (Recommendations for the authors):IntroductionLine 81: Why predict effects in DLPFC only? There could be a range of 'non-core' brain regions that support speech perception. Were the effects lateralized to left DLPFC predicted in advance of the study?

Indeed, as expanded upon in greater detail in Peelle (2018) and elsewhere, we agree there are many “non-core” regions that support speech perception in a variety of contexts. Briefly, our focus on DLPFC is motivated by (a) many fMRI studies, including from our group (not all published), showing DLPFC activity for degraded speech; (b) the work of other groups in the frameworks of cognitive control, multiple-demand, or frontoparietal attention networks implicating DLPFC in a variety of demanding tasks; (c) behavioral evidence for domain-general demands when processing acoustically-degraded speech, which we view as consistent with a role for DLPFC (as frequently characterized).

The lateralized DLPFC activity was predicted prior to starting data collection.

MethodsLine 91: How were the sample sizes of the CI group and the control group determined? I note that page 7 explains that 15-20 subjects per group was thought to be sufficient, based on similar studies. Did these previous studies include a cross-sectional design?

In these referenced studies (Defenderfer et al. 2021; Pollonini et al. 2014; Zhou et al. 2018b), the normal hearing cohort in an effortful listening/unintelligible speech situation was considered to be representing the CI user cohort.

Page 4. Table 1 reports left ear 4fPTA (unaided). The footnote for Table 1 reports that 8/20 CI users had a hearing aid but did they use them during the speech task?

Yes, all listeners performed the test in their everyday listening configuration.

Line 120. Where were the 2 speakers located relative to the participants? +/- 45 degrees? Which sound level were the words presented at?

We have added this to the text.

Lines 127-130. More details about the spoken word task is needed. Is this a standardized test or a novel test that the authors developed themselves?

This task was first introduced by Peterson et al. 1988 (Petersen, S. E., Fox, P. T., Posner, M. I., Mintun, M. and Raichle, M. E. Positron emission tomographic studies of the cortical anatomy of single-word processing. Nature 331, 585–589 (1988)), It was later replicated using HD-DOT by Eggebrecht et al. 2014, Sherafati et al. 2020, Fishel et al., 2020, in multiple studies to correctly localize the auditory cortex activation using both fMRI and HD-DOT.

Line 131. Which 'preliminary results'?

Clarified and changed to “To better understand the left PFC activity we observed in our first few CI users, we adopted a spatial working memory task introduced in previous studies”.

Line 191. "…based on the response … in a group of subjects…". Which subjects? Merge with relevant information on page 8.

Implemented.

Line 216. Is "Pollonini et al. 2014" cited correctly?

Yes.

Line 238. Why define only left DLPFC as an ROI?

In our experience with fMRI studies of acoustically-challenging speech, frontal activity seems qualitatively stronger and/or more common in the left hemisphere (see e.g., Davis and Johnsrude, 2003, Journal of Neuroscience).

Page 9. The visual spatial working memory task also activated 'auditory' regions in the right hemisphere that look like they overlap with the 'right auditory ROI' created by the authors. Do the authors consider this overlap to be problematic?

Our focus is on the left PFC region involved in spatial working memory. The observation of left temporal lobe activity is somewhat puzzling but does not affect our ROI definition.

Page 9. Legend for Figure 3. "…in the DLPFC region, survived after…". A word is missing.

Now fixed.

Page 10. Figure 4C suggests that CI > controls contrast identified quite a few group differences. So why then focus on left DLPFC?

Based on prior literature and our work in the area we had a specific hypothesis regarding left DLPFC. We certainly allow that there may be other areas involved in speech processing for CI users not engaged in controls, but that was not the focus of the current paper.

Page 11. Results of two-sample t-tests are reported in the legend of Figure 5. This information should be taken out of the legend and reported in full. Why were t-tests more appropriate than an ANOVA? Please report results of statistical analyses fully i.e. not just p-values. Were reported p-vals corrected for the number of t-tests that were used?

We have ensured the t statistics and degrees of freedom are in the revised manuscript. The ROI results in Figure 3 (main results) are Bonferroni corrected for multiple comparisons across the three ROIs tested, now also clarified in the text.

Page 11. The authors report the results of correlation analyses. Please add further details e.g. are Pearson's correlation coefficients reported?

Yes, Pearson correlations—now specified.

Page 11. The authors report that the threshold used for statistical significance in the correlation analyses was 'uncorrected'. Presumably this means that reported p-vals were not corrected for the 4 correlation analyses shown?

Correct.

Page 12. Correlation analyses for CI users. The rationale for these analyses is missing. Why carry out simple correlation analyses on data for CI users only? Potential predictors of speech processing (4PTA) could have been used as a covariate in the main analyses (e.g. Figure 5).

The CI listeners were our primary group of interest, and we also assumed they would show more variability in these covariate measures (supported by the data). We completely agree that some of these covariates might also be informative in the control listeners, but we worried there might be a confound (if controls and CI users differ both in group membership and 4PTA, for example, it would cause problems with interpretation). So, we opted to restrict these exploratory analyses to the patient group of interest.